# Assessing the Robustness of Ozone Chemical Regimes to Chemistry-Transport Model Configurations

Elsa Real [1,*], Florian Couvidat [1], Adrien Chantreux [1], Athanasios Megaritis [2,*], Giuseppe Valastro [2] and Augustin Colette [1]

[1] Institut National de l'Environnement Industriel et des RISques (INERIS), Parc Technologique ALATA, BP 2, 60550 Verneuil-en-Halatte, France; florian.couvidat@ineris.fr (F.C.); chantreux@ifca.unican.es (A.C.); augustin.colette@ineris.fr (A.C.)

[2] Concawe, Boulevard Du Souverain 165, B-1160 Brussels, Belgium; giuseppe.valastro@concawe.eu

[*] Correspondence: elsa.real@ineris.fr (E.R.); athanasios.megaritis@concawe.eu (A.M.)

**Abstract:** In a previous study, we assessed the efficiency of reducing either traffic or industrial emissions on various ozone metrics for several cities in Europe, based on the Air Control Toolbox surrogate model. Here, we perform various model parametrisation sensitivity analyses in order to assess the robustness of our results. We find that increasing the model resolution has a limited impact on the ozone response to emission changes when focusing on concentration peaks but strongly changes the response of the ozone daily mean with a switch to a titration regime for all zones with significant nitrogen oxide ($NO_x$) emissions. The impact of pollution imported from outside the simulation domain was also studied and we show that if the first lever for action on ozone peaks remains as the reduction of local and regional emissions, in order to achieve higher levels of reduction, it is necessary to act at a European level. We also explore more up-to-date temporal profiles and sectoral emission speciation and find a shift towards a more $NO_x$-limited regime in a number of cities. Overall, these sensitivity tests show that most of the differences are simulated in cities with high $NO_x$ emissions and little solar radiation but do not change the overall conclusions that were previously obtained.

**Keywords:** sensitivity analyses; ozone chemical regime; surrogate model

## 1. Introduction

Air pollution is a major health problem, particularly in countries experiencing rapid economic growth and urbanisation, such as India [1–3] and China [4]. In Europe, trends over the last 20 years [5] have tended to show a decline in pollutant emissions and concentrations. However, this decline is much less marked for ozone, and the trend even goes upwards for annual average ozone values. Ground-level ozone ($O_3$) is a harmful air pollutant known to affect morbidity and acute mortality ([6]) and to damage vegetation, affecting crops and forestry ([7]). Ozone is a secondary pollutant, which means that it is not emitted directly into the air. It occurs naturally in the Earth's upper atmosphere (stratosphere) and concentrations in the lower troposphere result from a balance between import from the stratosphere, chemical production and destruction, and deposition at the Earth's surface. Its chemical production results from chemical reactions between nitrogen oxides ($NO_2$ + NO) and volatile organic compounds (VOCs) in the presence of sunlight. Concentrations are more likely to reach values that are harmful to health on hot sunny days. In Europe, a north/south gradient is also observed in $O_3$ concentrations, with higher concentrations in southern regions.

The identification of management strategies to reduce ozone pollution is made considerably more complex by the fact that while $NO_x$ ($NO_2$ in this case) is a precursor of $O_3$, $O_3$ is also consumed by reaction with NO. Thus, in the presence of high concentrations of NO, $O_3$ concentrations can become very low. This elimination of $O_3$ by the reaction with

NO to form $NO_2$ is called titration. In the absence of NO, ozone is long-lived and can be transported over long distances in the atmosphere, affecting air quality in rural areas far from the source of the $NO_x$ emissions. In some cases, $O_3$ can even be transported on a continental or even intercontinental scale.

Another aspect that further complicates the study of the impact of ozone is that it was shown that, more than the daily average ozone levels, it is mainly temporal metrics that reflect daily peak ozone levels that are correlated with health effects [8]. Similarly, when it comes to the impact of ozone on vegetation, the threshold-based AOT40 indicator (Accumulated exposure Over a Threshold of 40 nmol $mol^{-1}$) was retained as the basis for the ozone critical levels for vegetation [9]. For these reasons, in addition to annual ozone mean concentration, the European Union (EU) has defined several standards, e.g., to characterise pollution episodes caused by ozone (information and alert threshold), to protect human health (long-term objective (LTO), i.e., the maximum daily 8 h mean concentration of ozone should not exceed 120 μg/m3 for more than 25 days) and to protect the vegetation (AOT 40 (Accumulated Ozone exposure over a Threshold of 40 ppb, expressed in $μg·m^{-3}·h$) is the sum of differences between hourly concentrations greater than 80 $μg·m^{-3}$ (=40 ppb) and 80 $μg·m^{-3}$ for a given period using the 1 h values measured daily between 8 am and 8 pm) and target value for vegetation (Directive 2008/50/EC). In addition, a specific metric is calculated to evaluate $O_{3's}$ impact on health ($SOMO_{35}$) ($SOMO_{35}$ (Sum Of Means Over 35 ppb, expressed in ppb·days) is the sum of max daily 8 h averages over 35 ppb (=70 $μg·m^{-3}$) calculated for all days in a year). Because of the intricate processes at play in the formation of ozone, the response of various ozone metrics (or indicators) differs when there are changing emissions in the main underlying precursor sources [10–12]. A study of trends in Europe since 2000 [5] showed an increase in annual mean ozone concentration while the high peaks were reduced. An increase in annual mean $O_3$ concentration is substantial at traffic sites (20%) and almost nil for rural ones. The increase in the mean level can be explained by the hemispheric transport of $O_3$ and the reduced titration by NO as a result of reduced $NO_x$ levels in the atmosphere. The clear difference between rural sites and other typologies indicates that the decreased titration has more impact on the recent trends in Europe than hemispheric transport. Because of the long-range transport impact and the highly non-linear chemistry of $O_3$, which differs according to emissions and meteorological conditions and, therefore, geographical areas, it is particularly complicated to understand, simulate and predict $O_3$ concentrations. To account for this complexity, chemistry-transport models are needed to simulate ozone concentrations. However, because they explicitly reproduce numerous chemical and physical relationships on each grid cell over a wide area, running these models is time-consuming and the number of air quality management strategy scenarios (i.e., quantity of emission reductions per sector) that can be tested is limited.

In a previous study ([13]), the surrogate model Air Control Toolbox (ACT) ([14]) was used to construct an Atlas of $O_3$ chemical regimes over 22 cities in Europe. Using this surrogate model allowed for the rapid exploration of the full range of reductions (0–100%) of $O_3$ precursors from different sectors while taking into account the non-linearity of $O_3$ chemistry. In that latter study, the changes in several ozone metrics as a result of reductions in road transport and industry emissions were evaluated. These two sectors were targeted because, in Europe, the road sector is the main anthropogenic sector emitting $NO_x$ and the industrial sector is the main emitter of NMVOCs and the third largest emitter of $NO_x$ [15]. The main conclusions were that:

(1) The $O_3$ sensitivity to road transport and industrial emission reductions differ from one city to another, but also for the same city when considering different ozone metrics (annual daily max vs. $SOMO_35$ for example) and from one period to another (summer vs. winter for example, or even different meteorological years);

(2) Counterproductive impacts on $O_3$ (i.e., increase in $O_3$ concentrations due to $NO_x$ emission reduction) are essentially a concern where and when ozone concentrations are low, typically below the current EU target value of 120 μg/m3. This is the case in winter,

for example, when ozone concentrations can more than double in some cities when traffic and industrial emissions are suppressed;

(3) Most cases show a higher $O_3$ sensitivity to emission reductions from road transport or equal sensitivity to emission reductions from road transport and industry. Very few cases are more sensitive to emission reductions from the industrial sector;

(4) Because of the importance of the natural ozone burden, the response of $O_3$ metrics to anthropogenic (industrial and road transport) emission reductions for the 22 cities analysed remains mostly at 37% for a 100% reduction of both industrial and traffic emissions when considering metrics averaged over a long period (summer, winter or annual average).

The present article is a follow-up of these conclusions. In particular, we aim to assess the robustness of our earlier findings on ozone mitigation strategies by further refining the modelling setup previously used. In air quality modelling research, many studies tackle the impact of model parameterisations, chemical schemes, emissions or resolution on absolute concentrations. Here, because we focus on ozone mitigation strategies, we are not just interested in the impact of modelled parametrisations on absolute ozone concentration, but above all, in the response of modelled ozone to reductions in precursor emissions. Such studies are not common. Most of them are rather model intercomparison exercises such as CityDelta [16] or Eurodelta [17]. In such studies, it is sometimes difficult to identify the parameter explaining the differences but they found that horizontal scale is an important factor that may change the $O_3$ response to emission changes, especially in cities due to the difference in reproducing the titration effect. They also found that low-scale models overestimate the impact of $NO_x$ and NMVOC reduction, compared to fine-scale models. Another study, [18], concentrated on evaluating the $O_3$ response to a 30% reduction of both $NO_x$ and non-methane volatile organic compounds (NMVOCs) for three sensitivity tests (i.e., increase anthropogenic NMVOCs by 40%, change in gas phase mechanism and change in vertical mixing). The authors found that the gross structure of the chemical regime ($NO_x$-sensitive and NMVOC-sensitive regime) mainly remained unchanged. On the other hand, they found that the chemical regime strongly responded to past or projected emission changes over Europe between 1980 and 2020 with a clear decadal tendency towards more $NO_x$-sensitive regimes over Europe. The impact of the grid resolution on the ozone response to emission reductions was also studied in [19]. For grid spacings ranging between 36, 12 and 4 km, they found that in urban areas, the coarsest (36 km) resolution fails to capture the extent and magnitude of VOC sensitivity and underpredicts the non-linearity of the ozone response indicated by the finest resolution. All these studies focus on the ozone response to a given emissions reduction (30% or a reduction relative to a forward-looking scenario). The aim of this paper is also to study the ozone response but for all possible reductions in ozone precursors (for the industrial and road sectors).

To explore the whole range of emission reduction impacts, a surrogate modelling approach is used. The ACT (Air Control Toolbox) surrogate model ([14]) is based on the full Chemistry-Transport Model (CTM) CHIMERE simulations. Because it is updated on a daily basis, it captures the meteorological variability well and was shown to capture the CHIMERE response with a limited error of 2%. Since that surrogate model is non-linear and includes interaction terms between various activity sectors, it is particularly suited to explore ozone chemical regimes in relation to reductions in both industrial and traffic emissions. New versions of the ACT model were created to test different parameterisations and configurations. First, we focus on the spatial resolution of the model by testing a new version with a finer horizontal grid resolution: 4 km $\times$ 4 km over South-East FRAnce, compared to an approximate resolution of 25 km $\times$ 25 km over Europe. In order to limit the computation time, this model has not been extended to the whole of Europe. South-east France was chosen because it includes two large cities (Marseille and Lyon), with different meteorological conditions, important industrial areas (Etang-de-Berre and Lyon industrial areas) and it is also regularly subjected to high ozone concentrations ([20]). In the second new version of the model, the impact of changes in anthropogenic NMVOC emissions and chemical mechanisms are tested. To achieve this, the different simulations based on

alternative NMVOC emissions and chemical schemes in the full CHIMERE model were first evaluated against the observations. The most relevant model setup was then used to build a new version of ACT and assess the impact on ozone responses to emission reductions.

The paper is presented as follows: First, in Section 2, the methodology is described. Then, Section 3 is dedicated to the results obtained over south-east France using the high-resolution (4 km × 4 km) version of the surrogate model. In Section 4, the relative importance of south-east France and inflow at the boundary of that region is assessed. Emissions parametrisations and chemical schemes are evaluated in Section 5. Finally, a discussion and conclusions are given in Section 6.

## 2. Methodology

### 2.1. The CHIMERE (CTM) Model

The air quality simulations used for both the design and everyday training of the ACT tool are performed with the CHIMERE Chemistry-Transport Model ([21,22]). The model is widely used for air quality research and applications ranging from short-term forecasting ([23]) to projection at climate scale ([24]). The CHIMERE v2020r1 is used in this study ([25]). Considering the setup used in the present study, the most important changes compared to the 2016 version used in [13] concern the biogenic and natural emissions with new parametrisation for sea salts and improved schemes for biomass burning emissions, mineral dusts and lightning as well as better consideration of biogenic VOCs in ozone chemistry.

The CHIMERE model simulates the transport and chemistry of atmospheric species in order to quantify the evolution of a plume of pollutants as a function of time on different domains (from urban to continental). From meteorological and emission flux data, CHIMERE allows the calculation of three-dimensional hourly fields of pollutant concentrations in the atmosphere. Because of the input data used, the number of equations to be solved and the physico-chemistry represented, CHIMERE is a mesoscale model, i.e., simulating the troposphere (from the Earth's surface to 200 hPa, i.e., an altitude of about 10 km) for a horizontal resolution of 1 to 100 km and for study domains ranging from cities to continents. The CHIMERE model simulates the formation and evolution of atmospheric particles ranging from a few nanometers to 10 μm. Aerosols in CHIMERE are composed of primary species, which are emitted directly by human activities, secondary inorganic species formed in the atmosphere such as sulphates, nitrates and ammonium as well as secondary organic species but also natural species such as sea salts and dust. The initial chemical mechanism implemented in CHIMERE is MELCHIOR2 ([26]). It is a simplification of the original MELCHIOR scheme ([27]) in order to limit the computation time and includes less than 70 species and around 120 reactions. Meteorological data are based on operational analyses of the IFS (integrated forecasting system) model of the European Centre for Medium-Range Weather Forecasts (ECMWF). The chemical boundary conditions for runs on European domain are obtained from ECMWF, also with the IFS model. Two resolutions were tested: 25 km × 25 km, and a zoom over south-east France with a resolution of 4 km × 4 km.

### 2.2. Emissions

The annual anthropogenic emissions in the reference simulations are taken from the CAMS-REG v3.1 inventory ([28]). This inventory is based on country report emissions from the Convention for Long-Range Transboundary Air Pollution and collected by the Centre for Emission Inventories and Projections (http://www.ceip.at/, 11 January 2024). Temporal factors are used to calculate hourly emissions from yearly emissions. In the base version of CHIMERE, emissions temporal profiles are taken from the GENEMIS project ([29]), except for traffic emissions, for which the temporal profiles of [30] are used. In order to be used within CHIMERE, total NMVOC emissions have to be split into CHIMERE model species. For this purpose, a speciation of NMVOC emissions must be performed. The standard CHIMERE NMVOC speciation is based on [31]. NMVOC are split into 23 different classes (alcohols, propane, butanes, etc.) The impact of changing both the temporal profile database

and the speciation of NMVOC emissions is evaluated in Section 5. Biogenic emissions are calculated on-line with CHIMERE using the MEGAN model ([32]).

### 2.3. The ACT Model

The Air Control Toolbox (ACT) was developed by INERIS as part of the Copernicus Atmosphere Monitoring Service (CAMS). ACT is a surrogate model that aims to reproduce the particles, nitrogen dioxide $NO_2$ and $O_3$ concentrations response of a CTM (CHIMERE in the case of ACT) to emission reductions from a specific sector. It is based on a polynomial function and trained on a dozen CTM sensitivity scenarios in which primary pollutant emissions are reduced. It is designed to be updated on a daily basis, i.e., the fitting of the parameters of the polynomial function is re-calculated every day based on the scenario CTM runs. ACT is able to reproduce the non-linearity in CTM response to changes in $NO_x$ and VOC emissions that are important for $O_3$. ACT is made available through a web interface (https://policy.atmosphere.copernicus.eu/CAMS_ACT.php, 11 January 2024) for the day-to-day forecast of the impact of emission reduction scenarios on air quality. As the purpose of this study is to assess the sensitivity of ACT to different schemes and parametrisations, new versions of the surrogate model were developed on the basis of alternative CHIMERE setups. Whereas the simulation of an emissions reduction scenario over a full year requires several days of simulation with a CTM, once calibrated, the ACT model enables us to obtain these results directly, without any simulation time. This makes it particularly interesting when studying the model's response to a whole spectrum of reductions from 0 to 100%.

For the time being, the model has been designed in such a way that emissions reductions only apply to the entire domain (Europe for the operational version) and over the long term (i.e., the emission reductions are assumed to be permanent, as opposed to short term measures that would apply for just the duration of an air quality episode). A full description of the ACT surrogate model design is given in [14], where it was demonstrated that it shows relative errors below 1% at 75% of the grid points and days, below 2% at 95% of the grid points and days, and below 10% for any grid points and days. ACT is configured to accept parametric emission changes in four activity sectors based on the SNAP categorisation. These are agriculture (AGR), industry (IND), residential heating (RH) and road transport (TRA). The exact SNAP sectors selected in each category are given in [14].

### 2.4. Numerical Experiment Description

Previous studies have investigated the sensitivity of ozone modelling predictions to different parameters and/or model input data. Among the most important factors are the resolution of the models the long-range transport, the meteorological parameters and the NMVOC and $NO_x$ emission rates and emission dynamical schemes [18,33–36]. Although in our study, we are not only looking at the sensitivity in terms of the modelled ozone but above all in terms of ozone response to an emission reduction, we carried out our sensitivity analysis on these main parameters: horizontal resolution of the model, boundary conditions at a regional scale and emission parameters (temporal profiles and sectoral emission speciation). For all these parameters, the full modelled ozone response to emission reduction from 0 to 100% (for traffic and industry) is tested. A last simulation was carried out on the chemical mechanism but did not result in a full sensitivity analysis of ozone response (see Section 5.1). The set of data characterising these tests is summarised in Table 1.

Two time periods are considered here: summer (JJA) 2019 for assessing the impact of increasing horizontal resolution and boundary conditions (Sections 3 and 4), with a focus on south-east France, and summer (JJA) 2018 for the one concerning the emission parametrisation impact (Section 5). The significance of the effect of the parameterisation change is then analysed specifically for several cities in the simulation domain. In each case, changes in several ozone indicators are compared for industrial and road emission

reductions from 0 to 100%. These comparisons are made in absolute and relative values, and also in the form of isopleths. Those specific cities are Lyon, Marseille, Fos-sur-mer, Bourgoin-Jallieu and OHP in south-east France for Sections 3 and 4; while the same cities as in 1 were studied in Section 5 (Lisbon, Sevilla, Madrid, Barcelona, Marseille, Fos-sur-mer, Paris, Milan, Roma, Antwerp, Brussels, Amsterdam, Copenhagen, Hamburg, Berlin, Prague, Warsaw, Beograd, Bucharest, Sophia, Athens and Nicosia).

**Table 1.** Description of model parametrisation sensitivity runs.

| | REF (EUR25 or Passant-GENEMIS) | SEFRA04 (or BC-EUREF) | BC-EURED | TNO-SPEC-TEMP |
|---|---|---|---|---|
| Annual Emission | CAMS-REG v3.1 inventory for year 2018 | CAMS-REG v3.1 inventory for year 2018 | CAMS-REG v3.1 inventory for year 2018 | CAMS-REG v3.1 inventory for year 2018 |
| Meteo | IFS (ECMWF) | IFS (ECMWF) | IFS (ECMWF) | IFS (ECMWF) |
| Domain | Europe | South-East France | South-East France | Europe |
| Horizontal resolution | $25 \times 25$ km | $4 \times 4$ km | $4 \times 4$ km | $25 \times 25$ km |
| Boundary Condition | IFS Global model | REF (EUR25) run with the same emission reductions | REF (EUR25) without emission reductions | IFS Global model |
| Chemical mechanism | Melchior 2 | Melchior 2 | Melchior 2 | Melchior 2 |
| Emission VOC speciation | Passant | Passant | Passant | TNO (CAMS-REG) |
| Temporal profiles | GENEMIS | GENEMIS | GENEMIS | TNO (CAMS-REG) |

## 3. Impact of Increasing Horizontal Resolution

### 3.1. ACT Model Design

An important shortcoming in the current operational ACT version operated in the Copernicus Atmosphere Monitoring Service lies in the moderate spatial resolution (0.25 degrees, about 25 km $\times$ 25 km), which does not capture urban processes well, in particular in relation to traffic $NO_x$ titration for ozone. To assess the impact of model resolution on the results of the ACT model, a high-resolution nested version of ACT is tested over a summer period (summer 2019: June, July and August), for south-east France. As for the online ACT model, this version relies on a dozen full CHIMERE sensitivity daily simulations but at a resolution of about 4 km $\times$ 4 km. We called this ACT version ACT-SEFRA04. When building the ACT model on a smaller domain, there are two possibilities concerning the boundary conditions to be used. In all cases, they are derived from CHIMERE simulations on a European domain which we will call EUR25 (for Europe at 25 km $\times$ 25 km). The first possibility is to use CHIMERE concentrations on the EUR25 domain for which the same reduction is applied on the large European domain EUR25 as on SEFRA04. This means that for each reduction scenario in SEFRA04 (e.g., 50% reduction of traffic emissions), the same reduction is applied over the European domain EUR25. This identical level of reduction is used for the dozen full CHIMERE simulations used to train the surrogate model ACT, and in turn, identical levels of reductions for the whole 0 to 100% range are available for both domains in the surrogate itself. In this case, the results obtained in the SEFRA04 domain are directly comparable to those obtained in the European EUR25 domain, since in both cases, the emission reductions were applied to the whole of Europe. We will call this ACT version BC-EURED (Boundary Conditions with REDuction over Europe). For simplicity of writing, when the ACT version is not specified, we considered by default that BC-EURED is the version used.

The second possibility is to keep emissions constant in Europe except for the emissions occurring within the SEFRA04 domain. In this case, emission reductions are only applied over SEFRA04. We will call this version BC-EUREF (European REFerence Boundary Conditions). The comparison between the two results (with a reduction on the whole domain or only on the SEFRA04 domain) allows us to study the impact of "regional"

reductions (i.e., on the scale of a large region) compared to reductions at the European scale (see Section 4). The simulation domain is represented in Figure 1 together with the number of exceedances of the maximum daily 8 h mean (MDA8) above the 120 μg/m$^3$ target threshold in 2019.

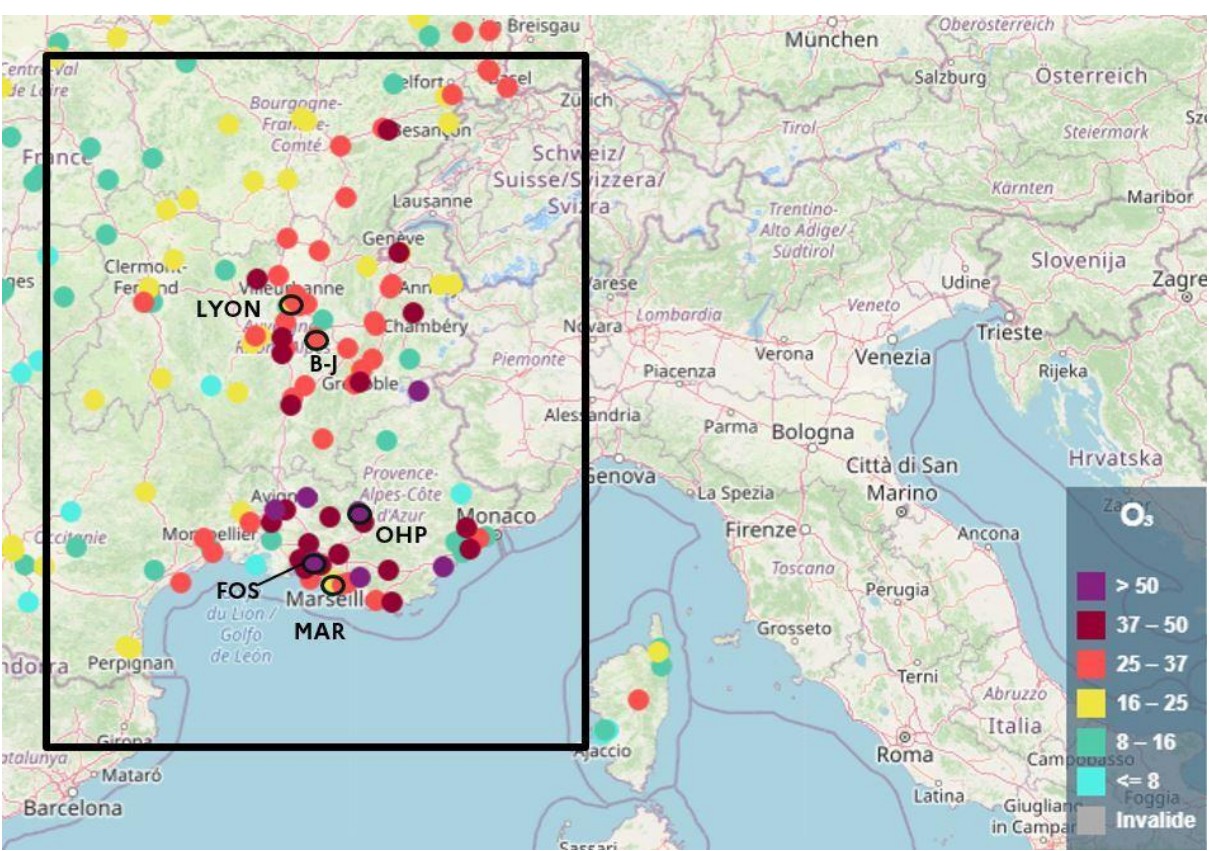

**Figure 1.** Annual number of O$_3$ exceedances of the 120 μg/m$^3$ threshold for the maximum daily eight- hour mean in 2019. Maximum number of exceedances for the EU's current target value is fixed at 25 days. This also corresponds to the percentile 93.15, which should not exceed 120 μg/m$^3$. The black rectangle characterises the SEFRA domain. The 5 stations used for analyses are highlighted with black circles.

For the present analysis, we focus primarily on five stations which we consider representative of different situations relative to O$_3$ in that region:

- Lyon (FR20062): The station itself is an urban station, rather influenced by traffic emissions but the area around Lyon is also known to be an area with high pollutant emissions from the industrial sector. Lyon is in the Rhône valley with a continental climate;
- Bourgoin-Jallieu (BJ—FR27007): This station is a suburban-type station. The emissions of ozone precursors are less important than in Lyon and because of the wind blowing from the Rhône valley towards the south, it is often found in the plume of pollutants coming from Lyon;
- Marseille (FR03043): The station itself is an urban station, rather influenced by traffic emissions. Emissions from the maritime sector are also important in this city. At the seaside, the influence of the sea breeze is important;
- Fos-sur-mer (FR02004): This station is located at the Etang-de-Berre, which is an important industrial area, mainly due to the refineries and petrochemical complexes located around the lagoon;
- OHP (Observatoire de Haute Provence) (FR24039): This station is of rural type, far from all important emission sources but regularly downstream of the plumes of the

Aix-Marseille area. Pollution plumes produce ozone when they move away from emission sources (and the ozone is no longer titrated by too much NO), a production also fueled by high levels of biogenic VOCs in the Provencal hinterland. As a result, significant levels of ozone are measured in OHP.

After an analysis of the performance of the full CHIMERE model in capturing $O_3$ concentrations and a comparison with observations, the main part of this section is dedicated to the analysis of the differences in $O_3$ response consecutive to road transport and industrial $O_3$ precursor emission reductions. Specific focus is given to the impact of emission reductions on daily peaks (Section Percentile 72.8 Results).

### 3.2. Impact of the Resolution on Modelled Ozone Concentrations and Evaluation against Observations

Figure 2 compares $O_3$ daily mean concentrations averaged over summer 2019 (JJA—June, July, August), as modelled using the two resolutions, together with the respective average $NO_2$ emissions. It should be noted here that the emission allocation method ensures that the total emissions over the study area are identical regardless of the resolution. The emissions are simply more diluted with the 25 km × 25 km resolution (upper left figure) with less pronounced emissions peaks in Lyon, Marseille or in the Rhône valley. Directly related to this spatial distribution of emissions, $O_3$ concentrations are found to be higher in the 25 km × 25 km resolution simulation compared to the 4 km × 4 km resolution in the vicinity of large $NO_x$ sources because of weakening titration in lower resolution models. Far from the large $NO_x$ emissions sources, the high-resolution simulation shows sharper $O_3$ gradients with more dissected high-concentration areas presenting stronger peaks. This can be explained by the lower dilution of $O_3$ precursor emissions in the high-resolution simulation, which form ozone when polluted plumes move away from areas of high emissions.

Table 2 compares the simulated daily maximum and daily mean $O_3$ concentrations to measurements over the SEFRA04 domain for all background stations (rural and urban). The results are also compared to those compiled in [37] from about 50 scientific papers that modelled $O_3$ concentration. Our results, either with the EUR25 or the SEFRA04 domain, are good, notably better than the mean scores compiled in [37].

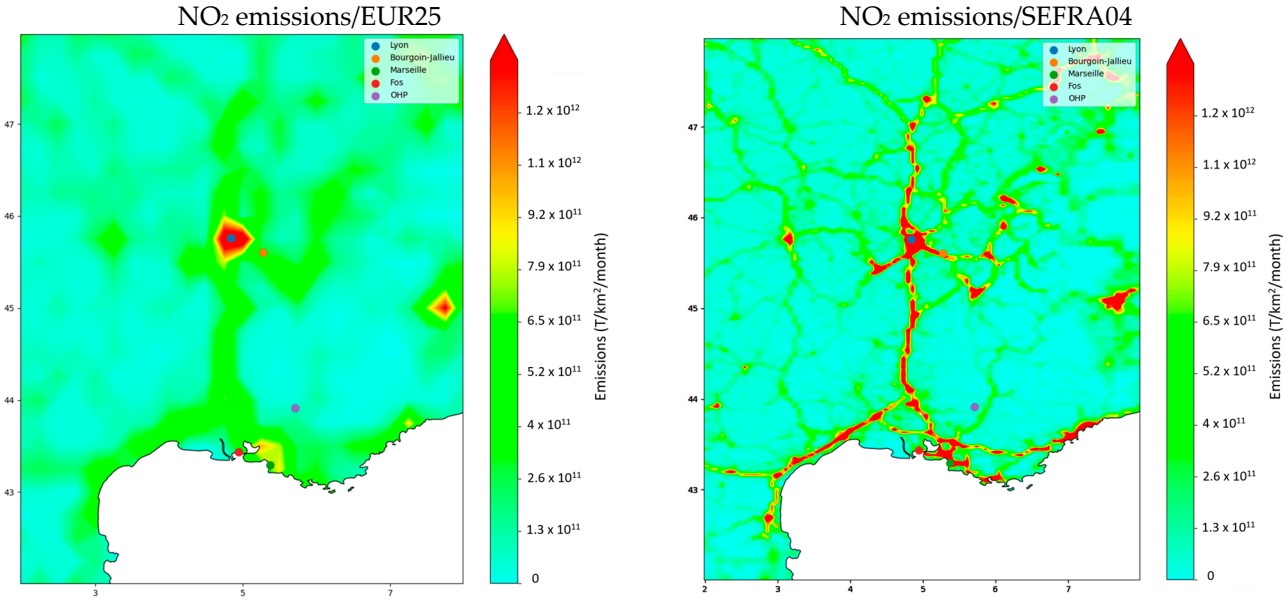

**Figure 2.** *Cont.*

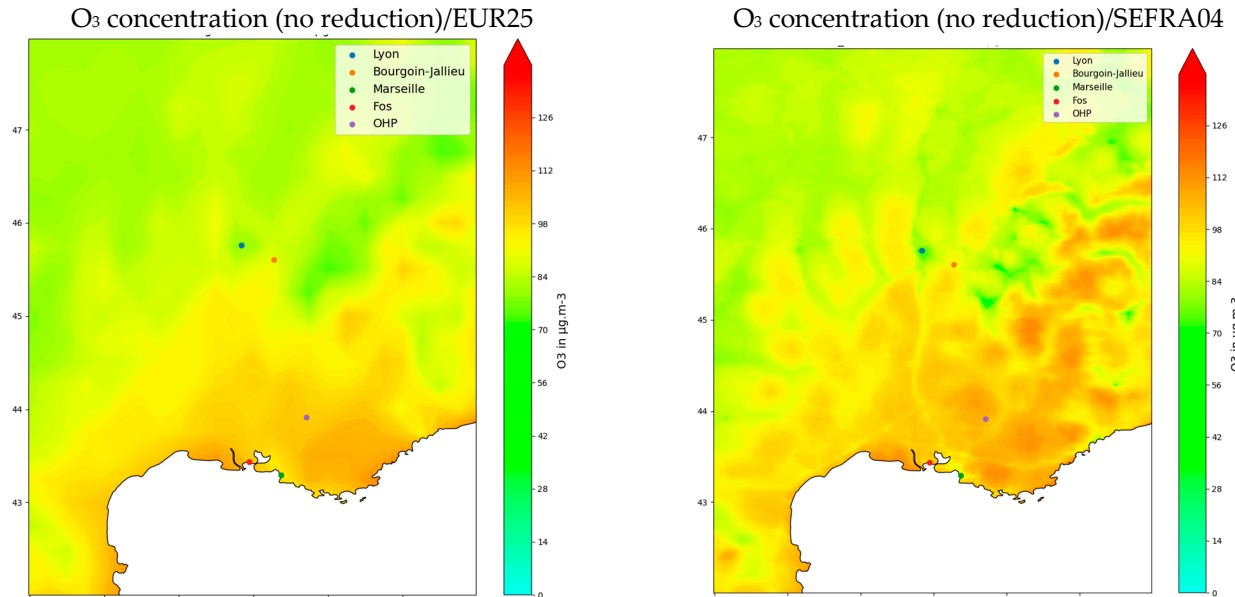

**Figure 2.** $NO_2$ average emissions in Tonnes/km$^2$ (**top**) and daily mean $O_3$ concentration (**bottom**) averaged over summer 2019 (JJA—June, July, August) simulated with no emission reductions for EUR25 resolution (25 km × 25 km) (**left**) and SEFRA04 resolution (4 km × 4 km) (**right**).

**Table 2.** $O_3$ daily max and $O_3$ daily mean correlation coefficient), bias and Root Mean Square RMSE (μg/m$^3$) averaged for all stations within the SEFRA04 region but using either the SEFRA04 or EUR25 spatial resolutions.

| | **$O_3$ Daily Max** | | |
|---|---|---|---|
| | Bias | R | RMSE |
| EUR25 | −3.8 | 0.79 | 18.53 |
| SEFRA | −1.0 | 0.76 | 18.43 |
| | **$O_3$ Daily Mean** | | |
| | Bias | R | RMSE |
| EUR25 | 5.9 | 0.77 | 23.9 |
| SEFRA | 4.9 | 0.75 | 24.0 |
| Sharma et al., 2017 ([37]) | 8.4 | 0.62 | 29.0 |

On average, $O_3$ daily maximum concentrations are higher when increasing the resolution. This leads to a significantly lower bias for all station types (rural, suburban and urban background) and RMSE is almost unchanged. In contrast, the correlation coefficient ®is slightly worse with the high-resolution simulation. The same statistics were calculated for the daily mean $O_3$ concentrations. Unlike $O_3$ daily maxima, which are underestimated, the model tends to overestimate the daily mean observed value. This positive bias is slightly reduced when increasing resolution, but the correlation coefficient is slightly worse and RMSE almost unchanged. This is in line with the conclusions from the literature review on $O_3$ modelling conducted in [37]. They showed that errors are reduced considerably, and various correlation metrics show improvement when researchers have used resolutions between 10 and 36 km compared to coarser resolutions. However, further enhancement in resolution does not necessarily result in improving model performance, especially for RMSE and R metrics, as also noted by [38].

### 3.3. Impact of the Model Resolution on Ozone Response to Emission Reductions

As was shown in [13], the ozone response to emission reductions in a given city can be very different depending on the ozone indicator: annual average, $SOMO_35$, daily max, percentile 93.15, etc. The analyses here, is also based on different types of ozone indicators, namely the summer average value of the $O_3$ daily mean concentration and the 72.8 percentile (P72.8) of the daily $O_3$ max values over the three summer months. The target value in the European Directive on Ambient Air Quality is defined so that MDA8 shall not exceed the EU target threshold of 120 µg/m$^3$ for more than 25 days in a given year. On an annual basis, this is the 93.15 percentile of MDA8 within the distribution of 365 values. Considering that most high MDA8 occur in summer, this would correspond roughly to the 72.8 percentile in the 92 days in June, July and August. Another simplification we had to adopt was to use the maximum of the daily hourly mean as an approximation for the MDA8, given that the ACT model only calculates the daily mean and the daily hourly max. Despite these approximations, the P72.8 indicator seems to be a good indicator of the expected general behaviour of the annual 93.15 percentile of MDA8. In [13], it was not possible to calculate $O_3$ daily average (only daily max) and this study provides an opportunity to compare the daily average and max $O_3$ values.

#### 3.3.1. Response to 60% Reduction in Traffic Emissions

Before exploring the whole range of emission reductions at the selected locations, it is relevant to consider a 60% reduction in emission over the whole domain to introduce the spatial variability of the response in ozone concentration. Therefore, we first discuss the ozone maps obtained for a uniform 60% reduction in traffic emissions as in Figure 3 (based on the summer average of the daily mean $O_3$ concentrations). This scenario has a strong impact on ozone mean concentrations with an important increase in $O_3$ in urban areas for the 4 km × 4 km resolution simulation (red areas) that are not present in the 25 km × 25 km simulation (white and light blue areas). This difference can be explained by a higher concentration of $NO_x$ in these urban areas for the 4 km × 4 km resolution simulation leading to a subsequent change in the chemical regime, which becomes more influenced by titration (as will be confirmed in Section 3.3.2).

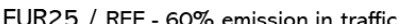
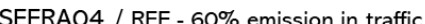
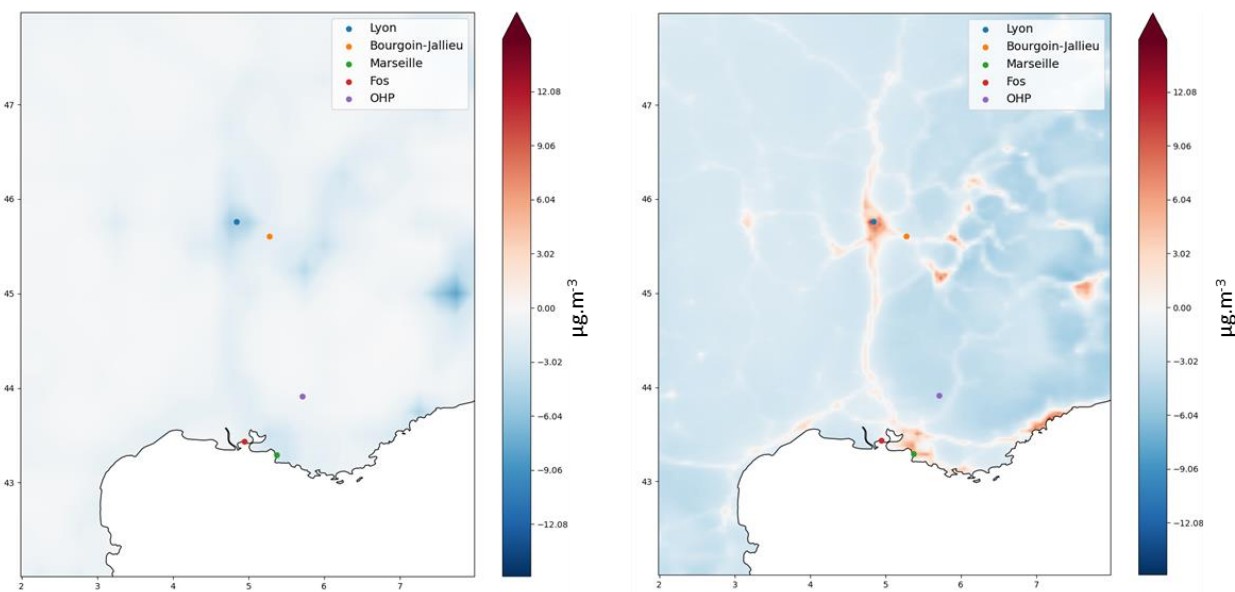

**Figure 3.** Difference in summer average daily mean $O_3$ concentrations (µg/m$^3$) between the reference simulation (no emission reductions) and a scenario where traffic emissions are reduced by 60% uniformly. Results for the EUR25 domain (25 km × 25 km resolution) are shown on the **left**, and for the SEFRA04 domain (4 km × 4 km resolution) in the **right** panel.

3.3.2. $O_3$ Response to the Whole Range of Emission Reductions

The added value of using the ACT surrogate model instead of a full CTM is that it gives us the opportunity to explore the response of ozone and the associated chemical regimes for the whole spectrum of reductions in road transport and industrial emissions (from 0 to 100% reduction).

In [13], this response was represented in two ways:

- Isopleths, based on the same principle as the [35] isopleths, but with the reductions in road transport and industrial emissions on the x-axis and y-axis instead of $NO_x$ and NMVOC concentrations (see also Section 2.3 in [13] for more details). For each city in the study, these isopleths were established for various $O_3$ indicators representative of the daily max ozone averaged over different periods (annual, summer, winter, sum of value >70 µg/m$^3$ to represent SOMO$_3$5 or percentile 93.15 to represent the EU target value);

- For a comparison of all cities, the total ozone changes for each day obtained by any combination of traffic and industrial emission reductions in a city were also represented by a boxplot.

On the basis of these results, six different ozone regimes were established: titration regime (complete or partial), more sensitive to traffic, to industrial or to both emission reductions, a net change in regime (from titration to ozone reduction) and a net change in sensitivity (see [13] and also Section 5.2).

The same representation of results is used here for three $O_3$ indicators: the percentile 72.8, the daily max value averaged over JJA 2019 and the daily mean value averaged over the same period. Isopleths and boxplots were established for the five locations described in Section 3.1.

Percentile 72.8 Results

For the sake of simplicity, isopleths are only shown (Figure 4: summer $O_3$ P72.8 response in % (negative changes, blue, for a decrease) corresponding to a given reduction in Traffic (TRA) and Industry (IND) emissions over Lyon (top panel), B-J (middle panel) and Fos-sur-mer (bottom panel). The results for the EUR25 resolution are shown on the left, and for the SEFRA04 resolution on the right. Both used BC-EURED (Boundary Conditions with REDuction over Europe). The absolute $O_3$ concentrations for the summer P72.8 in the reference simulation (when no reduction is applied) are displayed at the bottom left corner (in µg/m$^3$) (Figure 4) for the cities of Lyon, Bourgoin-Jailleu (B-J, often under the Lyon pollution plume) and Fos-sur-mer. The results for Marseille and OHP (often under the pollution plume of Marseille) are quite similar to those for Lyon and Bourgoin-Jailleu and are shown in the boxplot of Figure 5. The percentile 72.8 isopleths are compared for the two versions of ACT: high resolution (4 km × 4 km, SEFRA04 modelling domain) and low resolution (25 km × 25 km, EUR25 domain). In both cases, emission reductions are applied over the whole EU domain, which means that in the SEFRA04 modelling results, the boundary conditions used are those of the EUR25 simulation with the corresponding emission reductions.

The isopleths in Figure 4 show how the P72.8 (i.e., the 25th highest max value over the JJA months) changes (negative, blue, for a decrease) when reducing traffic (TRA) or industrial (IND) emissions between 0 and 100%. Starting from Lyon in the EUR25 simulation, in the reference simulations (without reducing either traffic or industrial emissions), the P72.8 value is 135 µg/m$^3$. Reducing either traffic or industry emissions leads to reductions of P72.8, while no adverse effect from titration is noticed. Because traffic $NO_x$ emissions are much higher than the respective emissions from industry in Lyon, reducing traffic emissions is more effective than reducing industrial emissions. A 100% reduction in traffic emission yields a 20% reduction (27 µg/m$^3$) in P72.8, while a 100% reduction in industrial emissions results in less than a 5% reduction (5 µg/m$^3$). The response also remains non-linear when removing both emission sources, which yields a 28% reduction (38 µg/m$^3$) as can be seen on the top right corner of the isopleth plot. Using the classification proposed

in [13], such an isopleth diagram would be classified as "traffic" sensitive. Using the higher resolution surrogate model for Lyon, the P72.8 value in the reference simulation (no emission reductions) does not differ a lot compared to the coarse resolution surrogate model (133 $\mu$g/m$^3$). The regime remains "traffic sensitive" but a very small titration effect is found (positive changes, red, for an increase) for emission reductions below 20%.

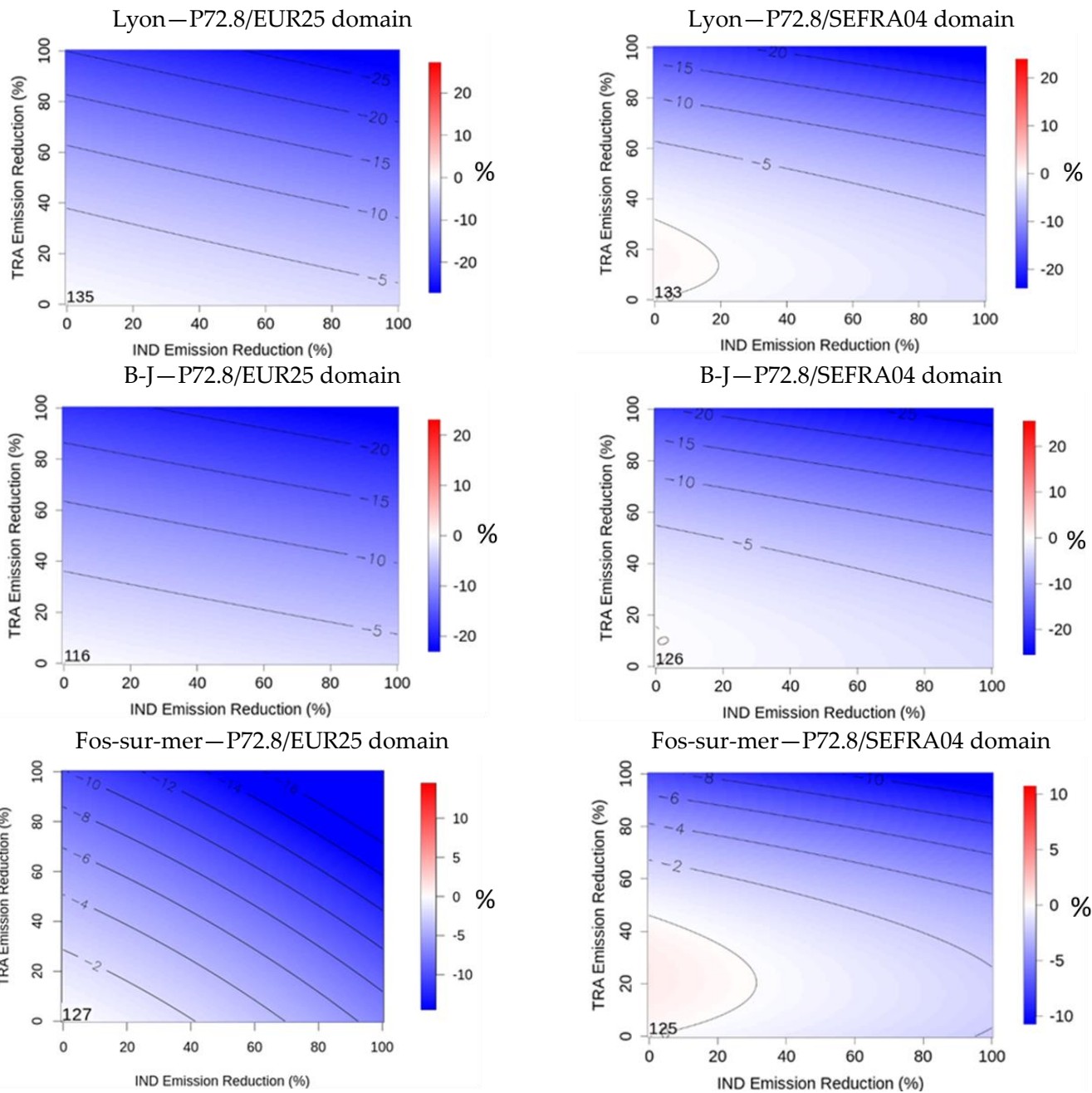

**Figure 4.** Summer O$_3$ P72.8 response in % (negative changes, blue, for a decrease) corresponding to a given reduction in Traffic (TRA) and Industry (IND) emissions over Lyon (**top** panel), B-J (**middle** panel) and Fos-sur-mer (**bottom** panel). Results for the EUR25 resolution are shown on the (**left**), and for the SEFRA04 resolution on the (**right**). Both used BC-EURED (Boundary Conditions with REDuction over Europe). The absolute O$_3$ concentrations for the summer P72.8 in the reference simulation (when no reduction is applied) are displayed at the bottom left corner (in $\mu$g/m$^3$).

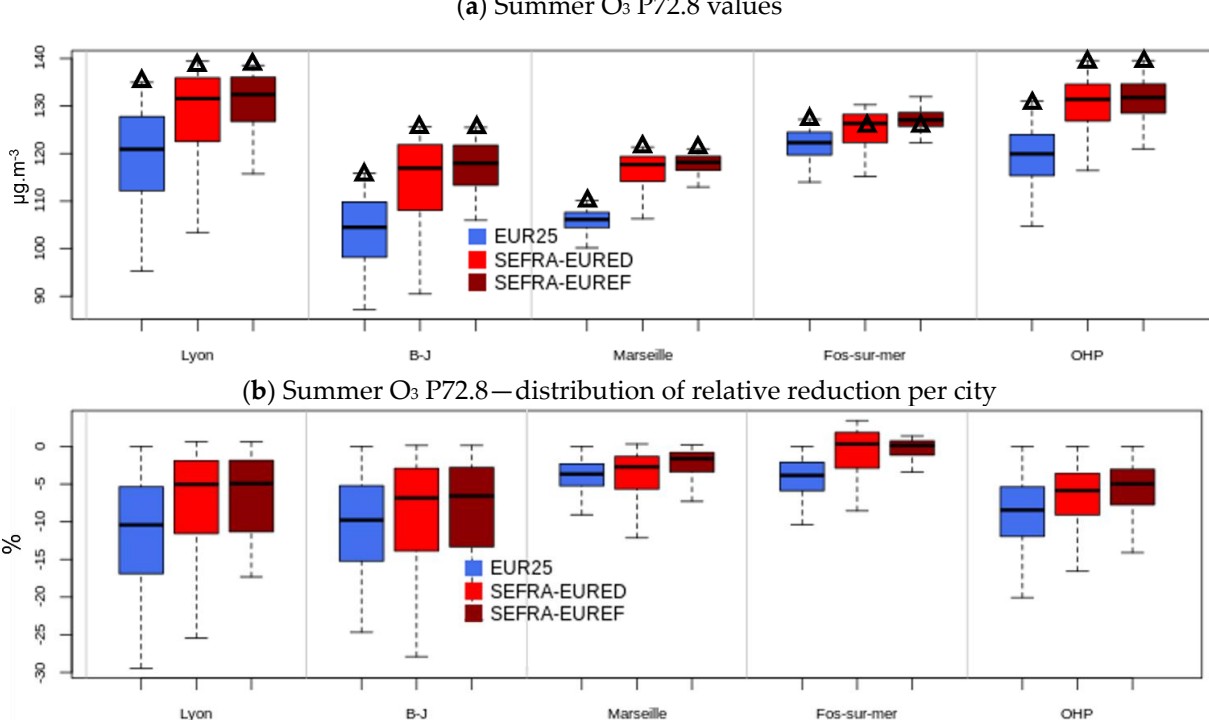

**Figure 5.** Distribution of summer $O_3$ P72.8 per city: Top (**a**): P72.8 values when emissions for both the industrial and the road transport sector are reduced between 0 and 100% by increments of 1%, and reference $O_3$ P72.8 value (triangle) where no reduction is applied. Bottom (**b**): distribution of $O_3$ P72.8 difference (in %) between the scenario where emissions are reduced and the reference without emission reductions. Blue: EUR25. Red: SEFRA04-EURED. Brown: SEFRA-EUREF.

When considering the B-J site in the background of the Lyon agglomeration, we also note a "traffic sensitive" regime. When moving to a higher resolution, the reference P72.8 increases from 116 $\mu g/m^3$ to 126 $\mu g/m^3$ but the regime remains unchanged. The reduction of industrial emissions in the SEFRA04 simulation yields slightly lower reductions in P72.8 than in the EUR25 simulation. For Lyon and B-J, the isopleths show a relatively similar ozone regime. It is in Fos-sur-mer that the change in the sensitivity is more pronounced. While in the EUR25 simulation, traffic and industrial emissions have a similar impact (which would be characterised as a "both industrial and traffic sensitive" regime), in the SEFRA04 domain, we move to a "traffic sensitive" regime and industrial emission reductions are less effective. We also notice a slight titration zone for low emission reductions. In Fos-sur-mer, both NMVOC and $NO_x$ emissions from the industrial sector are bigger than traffic ones but also more localised (concentrated on the main plants, as opposed to all roads for the road sector). The use of a higher resolution leads to an increase in industrial emissions close to their source resulting in increased ozone destruction due to high levels of NO (initial ozone is slightly lower: 125 $\mu g/m^{-3}$ vs. 127 $\mu g m^{-3}$). Because it is more localised and concerns high levels of emissions, the reduction in industrial emissions results in a small reduction in ozone (destruction and production offset each other). The reduction in road sector emissions in Fos-sur-mer but also in the surrounding areas then become more efficient in decreasing ozone concentrations.

Another way to look at the full range of $O_3$ changes is to analyse the boxplots in Figure 5 showing the distribution of $O_3$ values obtained for each city with traffic and industrial emission reductions ranging from 0 to 100% by increments of 1%. Here, each box shows the median, min and max as lines and a box between the 25% and the 75% quartiles. Boxplots with EUR25 (blue boxes) and SEFRA04 (red boxes) are compared in Figure 5. Brown boxes represent the values with EUREF boundary conditions that are discussed in Section 4.

This graphical representation is useful to refine the analysis because it shows on a single graph for the five cities, all the ozone indicator values taken when IND and TRA emissions are reduced from 0 to 100%. Figure 5a shows the indicator (here P72.8) absolute values as well as the initial ozone values (without emission reductions, represented as triangles), and Figure 5b shows the relative distribution (difference in % between the scenario where emissions are reduced and the reference without emission reductions) allowing to compare the $O_3$ responses.

Four types of behaviour can be observed:

- In Fos-sur-mer, increasing the resolution of the model leads to (1) a decrease in the initial simulated ozone concentrations (represented as a triangle) and (2) a reduction in the ozone response: reductions in traffic and industrial precursor emissions are less effective to reduce ozone 72.8 percentile (Figure 5b) and can even be counterproductive (positive relative difference, i.e., increase of ozone compared to initial values represented as a triangle). As explained earlier, high $NO_x$ emissions unaccompanied by high NMVOC levels leads to $O_3$ destruction which may not be offset by the increase in ozone production, resulting in a net $O_3$ destruction. A higher resolution leads to an increase in $NO_x$ near the sources and, therefore, an increase in $O_3$ destruction. In Fos-sur-mer, the increase in ozone destruction caused by the spatial concentration of $NO_x$ emissions results in both lower initial concentrations and lower ozone decreases when emissions are reduced.

- In Lyon, $NO_x$ emissions are more concentrated with a higher resolution. This leads to a small reduction in the ozone response when a higher resolution is used (for the same % of reduction, emission cuts are slightly less effective) but it does not result in lower initial concentrations.

- In Marseille and Bourgoin-Jailleu, the maximum reduction obtained for a 100% reduction in both traffic and industrial emissions is higher when increasing the resolution. Initial $O_3$ P72.8 values are also significantly higher with SEFRA04 by approximately 10 $\mu g/m^3$ in both areas. Here, when $NO_x$ emissions start to decrease, the same behaviour as for Lyon was simulated: a greater reduction in ozone destruction with the 4 km resolution (see B-J isopleths in Figure 4). However, after a certain reduction in $NO_x$, the behaviour reverses and a drop in ozone production (downwind production that is boosted with the 4 km resolution) dominates. This results in median reductions that are close to the two different resolutions, but strong cuts in emissions have a greater impact on ozone with the finest resolution allowing a greater maximum relative reduction (Figure 5b).

- In OHP, as in Marseille and B-J, increasing the resolution leads to a significant increase in $O_3$ initial value. In OHP, which is located in a rural area, high ozone levels are fed by pollution plumes from Marseille and the Etang-de-Berre (where Fos-sur-mer is located). Increasing the resolution has the effect of spreading these plumes less as they move around, therefore, favouring the production of ozone in rural areas. The extent of ozone reductions achieved by cutting emissions is slightly less with the higher resolution.

Summer Average of Daily $O_3$

The same analysis is performed in Figures 6 and 7 for the same period (JJA) but for the average of ozone daily mean concentrations over the period, instead of $O_3$ percentile P72.8 of the daily max. Fos-sur-mer, Lyon and Marseille show a similar behaviour (Figure 7): increasing resolution leads to (1) a decrease in initial $O_3$ value (black triangles), and (2) a switch to a titration regime (increased ozone when reducing emissions, i.e., positive reductions in Figure 7b). This switch toward a titration regime does not occur in Marseille and Lyon when considering ozone peaks through the P72.8 of $O_3$ daily max and this is linked to the daily ozone cycle. During the night-time there is no ozone production (because there is no solar radiation), so ozone is mainly consumed by NO. A decrease in NO concentrations thus leads to an increase in ozone at night, which is not seen when the indicator is the

daily maximum or the percentile 72.8. The 24 h average ozone behaviour relative to NO reduction depends on the balance between daily and nightly ozone evolution. In cities where NO emissions are high (which is the case of Lyon, Marseille and Fos with increased resolution), daily ozone increase is limited by the $O_3$ destruction by NO. In that case, the increase in ozone during the night dominates. And as increasing resolution also increases concentrations near sources, this explains the differences between EUR25 and SEFRA04. For OHP, which is located far from emission sources, $O_3$ has already been formed in the route and NO levels are relatively low. Therefore, the increase in ozone destruction at night with SEFRA04 is not predominant and the reference $O_3$ concentration is higher with SEFRA04. On the other hand, the ozone reductions achieved when reducing TRA and IND emissions are significantly greater with EUR25. B-J, which is more of a suburban area, is somewhere between a big city and a rural area. The increase in resolution does not lead to any titration phenomenon, but there is no increase in initial ozone either and the reduction associated with emission reductions is lower with SEFRA04.

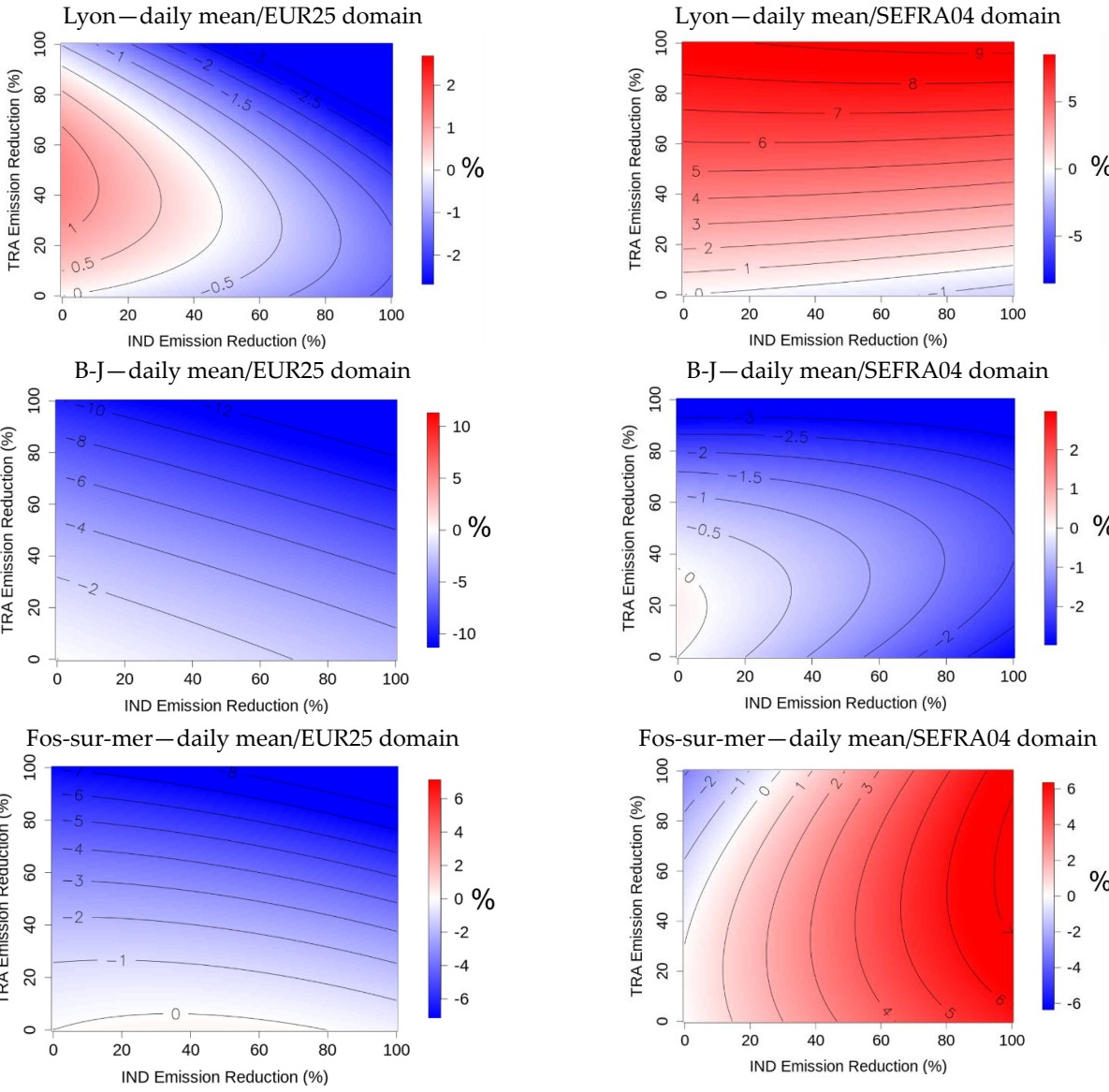

**Figure 6.** Summer average of the $O_3$ daily mean: response in % (negative for a decrease in blue) corresponding to a given reduction in Traffic and Industry emissions over Lyon (**top** panel), B-J (**middle** panel) and Fos-sur-mer (**bottom** panel). Results for the EUR25 resolution are shown on the (**left**), and for the SEFRA04 resolution on the (**right**). Both used BC-EURED (Boundary Conditions with REDuction over Europe).

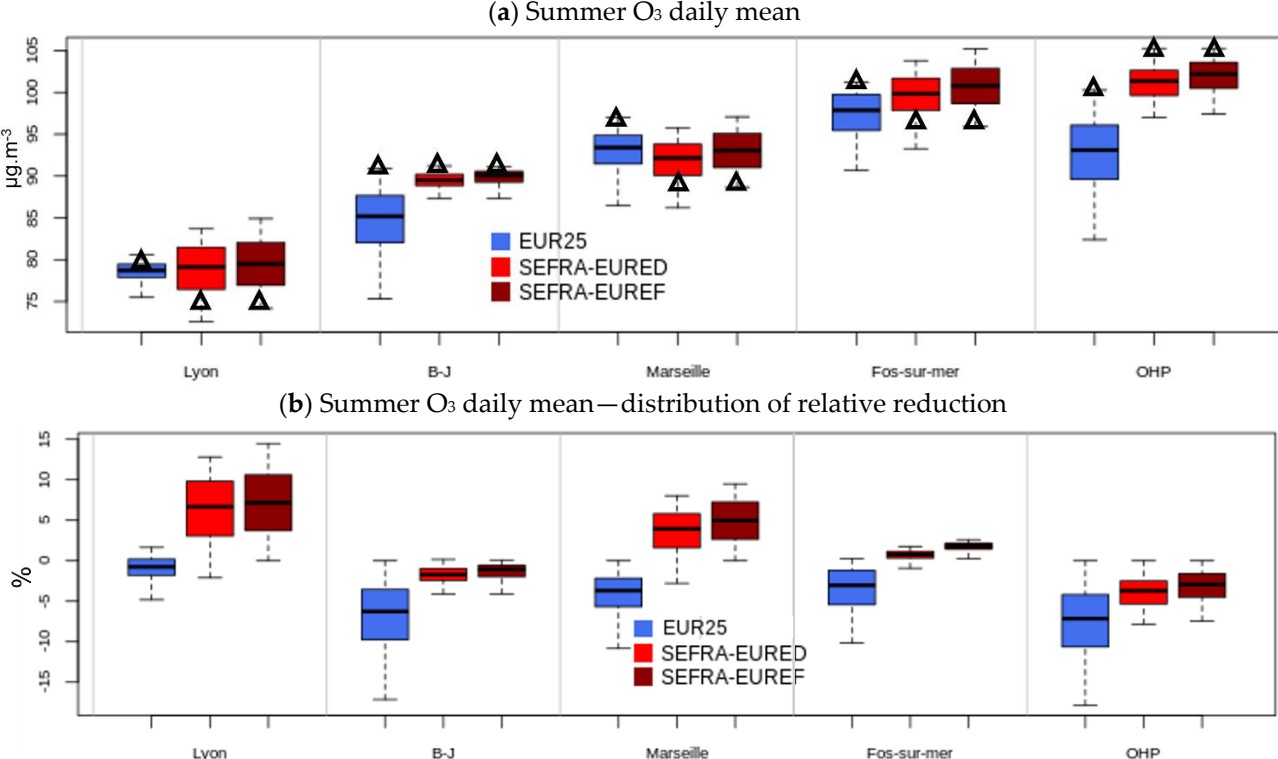

**Figure 7.** Distribution of the summer average of the $O_3$ daily mean per city: Top (**a**): $O_3$ daily mean values (in $\mu g/m^3$) when emissions for both the industrial and the road transport sector are reduced between 0 and 100% by increments of 1%, and reference $O_3$ value (triangle) where no reduction is applied. Bottom (**b**): distribution of $O_3$ daily mean difference (in %) between the scenario where emissions are reduced and the reference without emission reductions. Blue: EUR25. Red: SEFRA04-EURED. Brown: SEFRA-EUREF.

Thus, the impact of the change in resolution on chemical regimes is stronger for daily ozone averages, especially in areas with high ozone precursor emissions with a clear shift toward titration regimes. With regard to the amplitude of the ozone response (for the daily average indicator), it is lower with higher resolution for all types of localisations studied here (large city, suburban and rural areas). This confirms the results obtained in study [16] but with a wider window of precursor reduction. On the other hand, we observed that the results are different when the indicator is based on ozone maxima.

Focus on Ozone Peaks

In the previous sections, the analysis focused on values averaged over the whole period that is examined (summer). Here, the focus is on temporal trends and in particular on ozone daily peaks for two areas: the centre of Lyon and the entire SEFRA04 area. Figure 8 compares the time–series of $O_3$ daily max in an urban station in Lyon as measured (dotted line), simulated with the reference simulation (black line, no reduction) and for two scenarios for which emissions, either from traffic or from industry, are set to zero (TRA100%, blue line, and IND100%, green line, respectively).

It can be seen that in the reference simulation (black line), the temporal variability of the maximum concentrations is well reproduced. The peaks are well seen by the model but in general, they are underestimated. Conversely, low values are often overestimated by the model. This model behaviour is in line with the model score studied in Section 3.2 and is quite consistent with the performance of the chemistry-transport model as was seen in earlier studies ([37]).

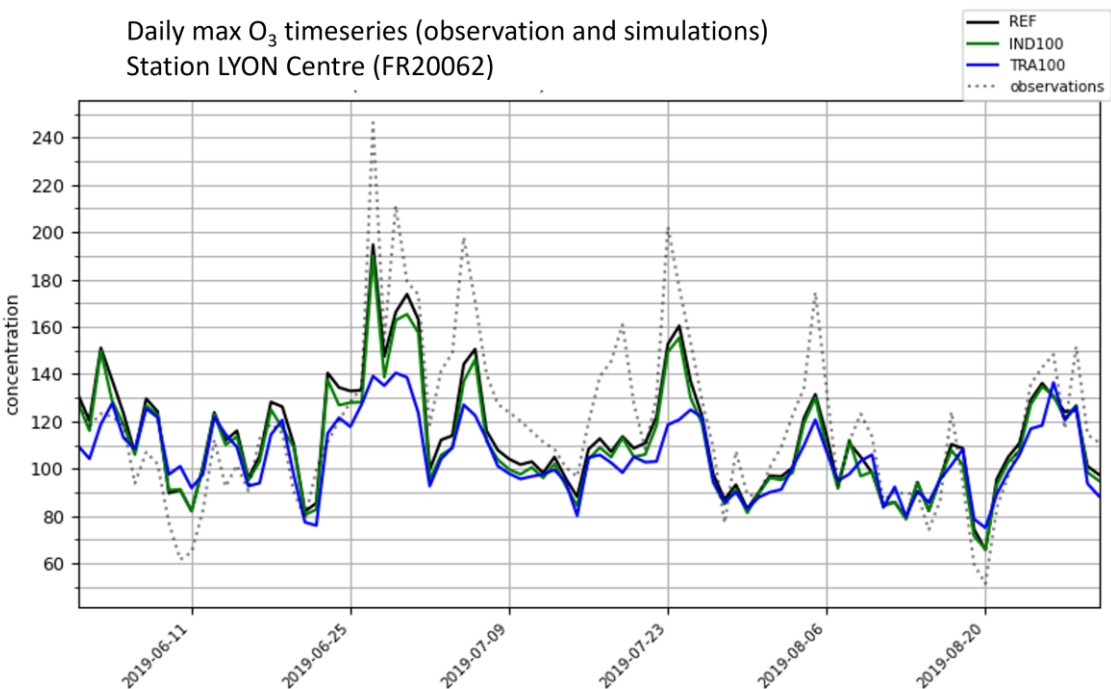

**Figure 8.** Time–series of $O_3$ daily max SEFRA04 (-EURED) simulation for the reference simulation (black line—no emission reductions), traffic 100% reduction (TRA100, blue line), industrial 100% reduction (IND100, green line) scenario and observations (dotted grey line) for Lyon station FR20062 during summer 2019.

In the centre of Lyon, industrial emissions do not have a strong impact on $O_3$ formation as reducing them by 100% (green line) only results in a few $\mu g/m^3$ decrease in $O_3$ daily max concentrations. On the other hand, a large concentration reduction is observed for the TRA100 scenario (blue line) for the highest peaks (e.g., end of June or end of July peaks). For the smaller $O_3$ concentration peaks, the effects from the reductions in the traffic scenario are less important, while an increase can even be observed for some days (e.g., 10th of August). The explanation lies mainly in the balance between ozone production resulting from the photolysis of $NO_2$ and ozone destruction by reaction with NO (titration effect). High ozone peaks inevitably mean that production is largely dominant. In this case, a reduction in $NO_x$ will lead to a reduction in ozone. When daily ozone peaks are lower, it is likely that there is (or getting closer to) a production/destruction equilibrium and the reduction in ozone destruction following that of $NO_x$ may dominate.

The analyses regarding daily peaks were also extended to the entire SEFRA04 region. In this case, the simulated number of days in exceedances of the AAQD information threshold (daily maximum above 180 $\mu g/m^3$) over the region was calculated (one cell showing a daily concentration above 180 $\mu g/m^3$ is accounted for one exceedance; a cell can be counted several times as being in exceedance). Exceedances of the 120 $\mu g/m^3$ threshold (EU target value) were also calculated and the evolution of these modelled exceedances as a function of the reduction in traffic and industrial emissions are presented in Figure 9.

Reducing traffic emissions significantly reduces the number of cells when the 180 $\mu g/m^3$ threshold is exceeded. A linear response is found for the reduction in exceedances up to 60% reductions in traffic emissions (which yield roughly 60% reductions in exceedances). Above 60%, the effectiveness is lower and with a 100% reduction in traffic emissions, there are still cells with exceedances (representing 13% of the cells initially in exceedances). Reducing industrial emissions is less effective, but still has an impact when the entire south-east region is considered. A 50% reduction in industrial emissions would reduce exceedances in south-east France by 30%. When a lower threshold is considered, such as the 120 $\mu g/m^3$ threshold, reductions in precursors are less effective with a maximum

reduction of 46% in exceedances when traffic emissions are completely suppressed. Once again, this can be explained by (1) a background level of ozone that cannot be lowered by reductions in anthropogenic emissions in Europe; (2) precursor emissions by other sectors (the solvent sector in particular for NMVOCs) and (3) the balance between ozone production and destruction.

(**a**) 180 μg/m³ threshold                                    (**b**) 120 μg/m³ threshold

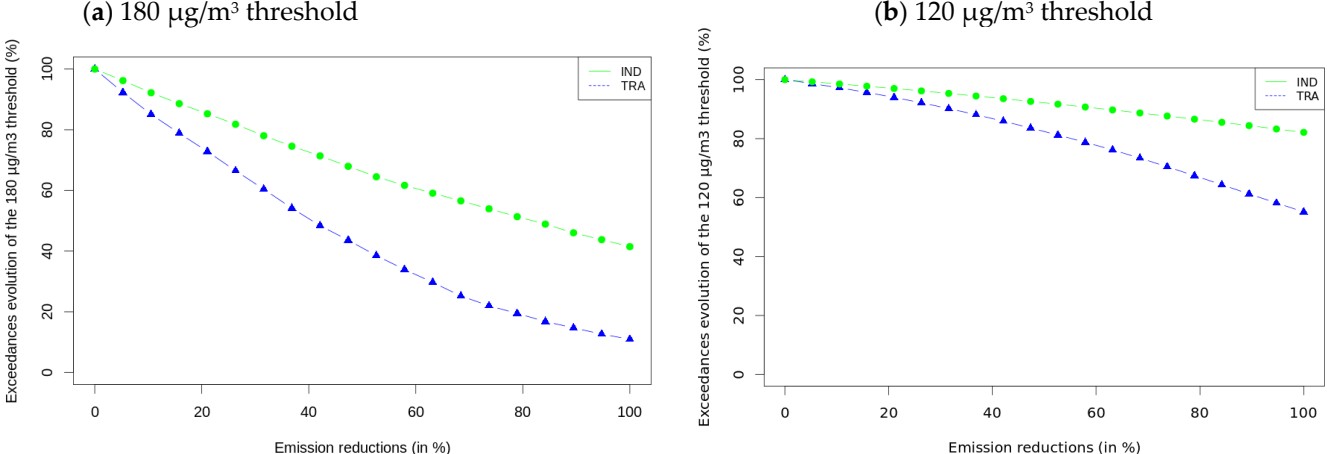

**Figure 9.** Evolution of the number of grid cells of the SEFRA04 region in exceedances of the (**a**) 180 μg/m³ and (**b**) 120 μg/m³ thresholds, respectively, as simulated with emission reduction from the Traffic (TRA) sector (blue line) and the Industrial (IND) sector (green line) ranging from 0 to 100%.

Over the south-east France region, the effectiveness of reducing industrial emissions in reducing ozone threshold exceedances is about half as effective as reducing traffic emissions.

## 4. Impact of the Boundary Conditions (BC-EUREF vs. BC-EURED)

The predicted $O_3$ concentrations and associated chemical regimes when different boundary conditions are used in the simulation domain were also analysed in this study. In one case, the emission reductions applied in the domain (SEFRA04) are also applied in the European domain that is used to calculate the boundary concentrations (i.e., BC-EURED for European REDuction). This is the case for all the analyses carried out and presented in the previous sections. In the other case, these boundary conditions are fixed, equal to the concentrations of the reference simulation (i.e., BC-EUREF for REFerence), so that emission reductions are only applied within the SEFRA04 domain.

The impact of changing boundary conditions (BC) on any emission reductions scenario can be analysed in Figures 5 and 6 by comparing the red (BC-EURED) and brown (BC-EUREF) boxplots. Only small differences are simulated for the JJA ozone daily mean average indicators (Figure 6) for all five cities. As already seen, this indicator is strongly influenced by $NO_2$ local emissions that cause titration and low ozone concentration, particularly at night. Concerning ozone daily peaks (P72.8, Figure 5), for emission reductions between 0 and 50%, the impact of limiting reductions to the region or to apply them over all Europe is low. This can be concluded from the similarity of the upper half of the boxplots related to the BC-EURED and BC-EURED simulations, at least up to the median values that are similar. On the other hand, very strong emission reductions will have an impact on long-range ozone transport, and simulations with emission reductions >90% result in an almost two-fold reduction in ozone peaks (min values of the boxplot) compared to simulations with only local emission reductions. This means that the first lever for action on ozone peaks remains the reduction of local and regional emissions, but that in order to achieve higher levels of reduction, it is necessary to act at European level to reduce ozone imports. This also means that ozone import can only be effectively reduced by high levels of emission reduction (>70%).

These conclusions would of course be different for a much smaller area where the import of ozone precursors would be more important leading to a higher difference between EURED and EUREF.

## 5. Impact of Changing Model and Emissions Parametrisation on the Ozone Response to Emission Reductions

In this section, we explore whether the simulated ozone response to emission reductions can be influenced by the chemical scheme used in the model, in particular the representation of anthropogenic VOCs in the chemical mechanism, but also the speciation of emissions per emission sectors and their temporal variability. This section is thus dedicated to the evaluation of the impact of these model modifications (chemical scheme, emission speciation or temporalisation) on ozone response and ozone chemical regimes. As in Section 3, the response is assessed using the surrogate ACT framework, with a surrogate model that is itself calibrated on alternative configurations of the full chemistry-transport model CHIMERE.

Updating ACT with either a new scheme or new emission treatment requires the repetition of the calibration process, meaning that 12 full CHIMERE scenario simulations are required. Before updating the ACT surrogate, a first step was carried out by analysing the concentrations modelled with the full-CHIMERE model. Two chemical schemes were compared: SAPRC-07 ([39]) and the original scheme in the CHIMERE model: MELCHIOR2 ([26]). The MELCHIOR2 and SAPRC-07 mechanisms are reduced implicit mechanisms and both use the lumping approach in order to gather into a single model several species with similar properties. The MELCHIOR2 mechanism includes less than 70 species and around 120 reactions. SAPRC-07 includes 85 species and 275 reactions and it is the most recent mechanism available in CHIMERE.

We also test the sensitivity to the VOC splitting used to distribute the total NMVOC emissions into individual model species. Provided using the CAMS-REG-V4.2 emission dataset ([40]), a new VOC speciation was used and compared with the original CHIMERE VOC speciation and temporal profiles (Passant speciation ([31]) and GENEMIS profiles, see Section 2.2). In this new VOC speciation, NMVOCs are split into 23 different classes (alcohols, propane, butanes, etc.) ([28]). We also explore alternative temporal profiles for emissions largely based on the earlier work from TNO (Nederlandse Organisatie voor Toegepast Natuurwetenschappelijk Onderzoek) ([41]). Because some VOCs are largely involved in ozone production, and because changes in the seasonal timing of $NO_x$ and VOC emissions can have a significant impact on ozone production (particularly in summer), it seemed important to study the impact of these new patterns on the ozone response to emission reductions. We will call these two parameterisations TNO-SPEC and TNO-TEMP, respectively. Both are dependent on the emissions sector.

### 5.1. Impact on Modelled Ozone Concentrations in the Reference Simulation

CHIMERE simulations with the different parameterisations, as described in Table 2, were conducted over the period from June 2018 to August 2018. Besides these combinations of parametrisations, the model set-up was identical to the one used in the previous sections. And the domain covered Europe with a resolution of 25 km × 25 km (EUR25 domain). Five (5) sensitivity runs were conducted with CHIMERE implementing either the TNO emission speciation (TNO-SPEC), the TNO emission temporalisation (TNO-TEMP), the SAPRC chemical scheme or a combination of them. The resulting ozone concentrations are shown in Figure 10.

The implementation of SAPRC (Figure 10b) leads to a decrease in ozone maxima with a variable response across EU. The implementation of more up-to-date temporal profiles and sectoral emission speciation (TNO-SPEC-TEMP) (Table 3) systematically leads to increased summertime ozone maxima across Europe (Figure 10c) with a maximum of 7% on average over the summer. These $O_3$ increases are mainly due to an increase in NMVOC concentrations across all European countries, reaching a maximum increase in Germany of

more than 40%, while $NO_x$ concentrations decreased slightly. The reason for these changes lies mainly in the time profiles used. Indeed, the new monthly profiles result in higher NMVOC emissions in summer (JJA), and slightly lower $NO_x$ emissions for the same period (see Appendix A).

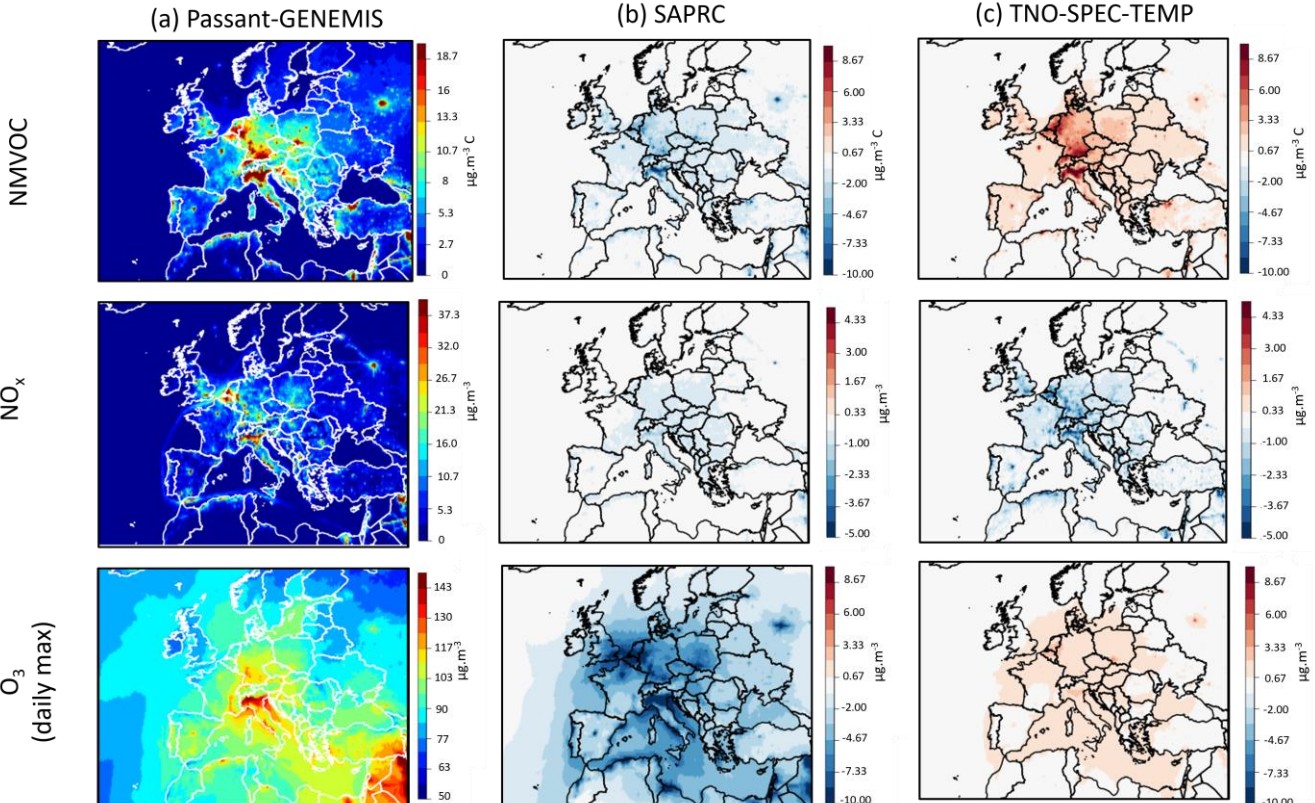

**Figure 10.** (**a**) Reference NMVOC, $NO_x$ and $O_3$ concentrations (μg/m$^3$) (Passant-GENEMIS) and absolute difference between (**b**) SAPRC and Passant-GENEMIS concentrations and (**c**) TNO-SPEC-TEMP and Passant-GENEMIS concentrations. Concentrations are averaged over the period of June 2018–August 2018 for daily mean values (NMVOC and $NO_x$) and daily max values ($O_3$).

**Table 3.** Modelling features of the CHIMERE sensitivity runs.

|  | Chemical Mechanism | VOC Speciation | Temporal Profiles |
|---|---|---|---|
| Passant-GENEMIS | Melchior 2 ([26]) | Passant ([31]) | GENEMIS ([29]) |
| TNO-SPEC | Melchior 2 ([26]) | TNO (CAMS-REG) | GENEMIS ([29]) |
| TNO-TEMP | Melchior 2 ([26]) | Passant ([31]) | TNO (CAMS-REG) |
| TNO-SPEC-TEMP | Melchior 2 ([26]) | TNO (CAMS-REG) | TNO (CAMS-REG) |
| SAPRC | SAPRC ([39]) | Passant ([31]) | GENEMIS ([29]) |

Another way to see the dispersion of concentrations and to evaluate the performance of the simulations is to compare them with observations (validated data from EEA AQ-reporting for background station (https://www.eea.europa.eu/data-and-maps/data/aqereporting-9/aq-ereporting-products, 11 January 2024)). The scores (RMSE, bias and correlation coefficient) averaged over the summer period (JJA) are reported in Table 4.

As the ozone concentrations are underestimated in Passant-GENEMIS, the simulation with the best score turns out to be the simulation combining TNO speciation and temporalisation and keeping the Melchior2 chemical scheme (TNO-SPEC-TEMP). In this case, the bias is reduced by a little more than 2 μg/m$^3$, as well as the RMSE. The scores of the SAPRC simulation are close to the Passant-GENEMIS simulation, although slightly worse.

**Table 4.** Model performance calculated on ozone daily maximum for the period JJA 2018 over Europe.

|  | Passant-GENEMIS | TNO-SPEC-TEMP | TNO-SPEC | TNO-TEMP | SAPRC |
|---|---|---|---|---|---|
| Mean Bias | −8.70 | −6.09 | −7.73 | −7.25 | −9.39 |
| R$^2$ | 0.78 | 0.78 | 0.78 | 0.77 | 0.77 |
| RMSE | 19.87 | 18.61 | 19.34 | 19.15 | 20.14 |

The model performances against observations and the analysis of the maps point in the same direction: the largest differences with Passant-GENEMIS are obtained with the TNO-SPEC-TEMP simulation and it is also the simulation that reproduces best the observed concentrations. For these two reasons, we decided to keep this version of the model to calibrate a new version of ACT. Because each ACT calibration requires 12 full CHIMERE scenario runs and regarding the small differences between the sensitivity runs, we decided to limit our study to one ACT calibration with that CHIMERE version.

*5.2. Impact on Ozone Response to Emission Reductions*

In this section, the impact of implementing the TNO-SPEC-TEMP emission parameterisation on the O$_3$ responses to emission reductions is studied over Europe, and in particular over the 22 cities that were included in [13]. For the sake of synthesis, the results are shown as boxplots for all the cities studied. As a reminder, each boxplot characterises the values taken by the ozone indicator when emissions (industrial and road) are reduced from 0 to 100% in increments of 1% (median, min, max as lines and a box between the 25% and the 75% quantiles).

Figure 11 presents all the ozone concentrations modelled when emissions are reduced (Figure 11a) but also the reductions in relative values in order to facilitate the comparison (Figure 11b). Regarding the change in initial ozone concentrations (i.e., without emission reductions) and by comparing triangles in the simulation with the original setup (Passant-GENEMIS) and for the TNO-SPEC-TEMP simulation (Figure 11a), there are three different cases:

(1) Cities where ozone values increase with the TNO-SPEC-TEMP parameterisation: Paris, Antwerp, Copenhagen, Brussels, Warsaw, Hamburg, Berlin, Prague, Bucharest and to a lesser extent Milan, Madrid and Amsterdam. Most of these cities have a NO$_x$/NMVOC emission ratio greater than 1.2 (see Figure 14 in [13]) and low O$_3$ values due to low radiation: these are NMVOC-limited cities: O$_3$ production is more sensitive to an increase in NMVOC, and even NO$_x$-saturated for the northern cities (titration). This explains the increase in O$_3$ due to TNO emission parametrisation implementation: the decrease in NO reduces O$_3$ destruction and the increase in NMVOC has a significant impact. Madrid and Milan are in a class of their own. These two cities have a fairly balanced NO$_x$/NMVOC ratio, with high concentrations of both NO$_x$ and NMVOCs. Reducing NO$_x$ and increasing NMVOCs, therefore, have a relatively low impact in favour of an increase in net ozone production.

(2) Cities where the implementation of TNO parametrisations has led to a decrease in ozone values (Rome, Fos-sur-mer, Barcelona, Marseille, Nicosia). These cities have also a balanced NO$_x$/NMVOC ratio and are all Mediterranean cities with high O$_3$ values. The reduction in NO$_x$ causes a greater drop in ozone production than the increase induced by the rise in NMVOCs. The Fos-sur-mer case is slightly different as it presents a higher NO$_x$/NMVOC ratio and should be more NMVOC-sensitive as the cities cited in previous case 1. However, it is also the only southern city in this NMVOC-limited regime, meaning

that photochemical reactions are more important. In that case, the decrease in $O_3$ production caused by lower $NO_x$ dominates the increase in $O_3$ following the NMVOC rise.

(3) Cities that show almost no differences (Sofia, Athens, Belgrade, Lisbon, Sevilla) for which the phenomenon in favour of an increase in $O_3$ concentration (NMVOC rise and $O_3$ destruction decrease due to NO reduction) are balanced by a phenomenon that leads to a fall in $O_3$ production (decrease in $NO_x$ concentration).

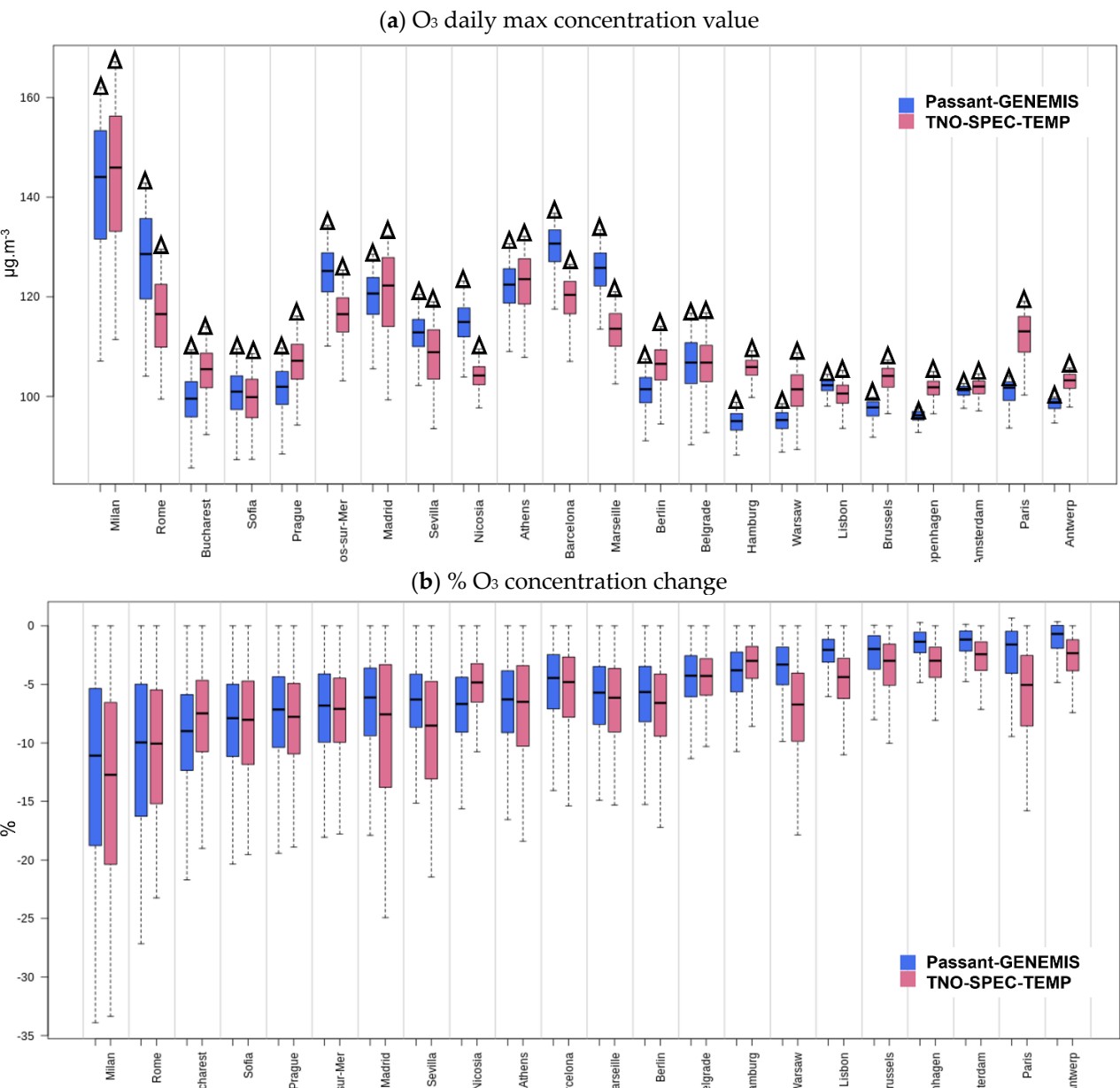

**Figure 11.** Distribution of $O_3$ daily maxima over the summer 2018 for 22 cities in Europe. Top (**a**): absolute concentrations when emissions for both the industrial and the road transport sector are reduced between 0 and 100% by increments of 1%, and reference $O_3$ value (triangle) where no reduction is applied. Bottom (**b**): distribution of the concentration difference (in %) between the scenario where emissions are reduced and the reference without emission reductions. Blue: Passant-GENEMIS, pink: TNO-SPEC-TEMP. Negative values mean reduction in $O_3$ consecutive to emission reductions.

After assessing the impact of the change in parameterisation on the initial ozone, the impact on the response of ozone to reductions in emissions was investigated. This is

better seen on Figure 11b. The comparison of the boxplots is quite revealing: For cities showing the largest ozone reductions, the magnitude of the response associated with the emission reductions (average and quartiles) is roughly the same in the two versions. For cities whose ozone concentrations are less responsive to emission reductions (right part of the figure: Warsaw, Lisbon, Paris, Brussels, Copenhagen, Amsterdam, Antwerp) the changes are more significant: the average ozone reduction is doubled with the new version, as is the maximum quantile. As we have already seen, the vast majority of these cities have a $NO_x$/NMVOC emission ratio greater than 1.2 and are, therefore, more likely to be NMVOC-limited cities. The general increase in NMVOC emissions in summer with the TNO-SPEC-TEMP simulations causes a shift towards a more balanced regime. As a result, $NO_x$ reductions become more effective for these cities.

The classification used to characterise the chemical regimes obtained for the different indicators and time periods chosen in [13] is also used here. For each of the 22 cities, and for the summer average of daily maximum ozone, the isopleths were classified into six different $O_3$ classes in terms of chemical regimes:

- Titration regime (complete or partial): reductions in emissions (IND or TRA or both) lead to an increase in the summer average of ozone daily maxima for more than half of the emission reduction pairs. This can be the case for any reduction (complete titration regime) or only for some part of the IND:TRA reduction space (partial titration regime);
- TRA sensitive: reductions in road transport emissions produce a greater reduction in the summer average of ozone daily maxima than for reductions in industrial emissions;
- IND sensitive: reductions in industrial emissions produce a greater reduction in the summer average of ozone daily maxima than reductions in road transport emissions;
- TRA and IND sensitive: road transport and industrial emission reductions have a similar impact on the summer average of ozone daily maxima;
- Change in regime: a decrease in $O_3$ metrics is the dominant reaction to emission decreases but an increase in the summer average of ozone daily maxima occurs in a small part of the IND:TRA reduction space;
- Change in sensitivity: There is a clear shift from a regime sensitive to road transport emission reductions to a regime sensitive to industrial emission reductions (or the reverse). This case was not encountered in the cities and over the period selected.

Figure 12 compares the frequency distribution obtained for the 22 cities in the summer of 2018 with an ACT surrogate model trained on the CHIMERE setup based on Passant-GENEMIS or on TNO-SPEC-TEMP.

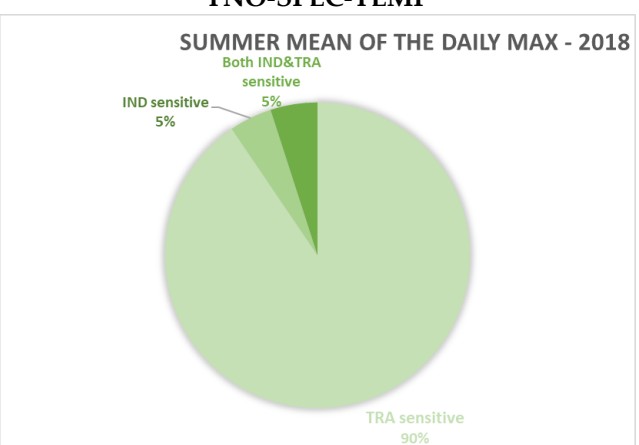

**Figure 12.** Classification of ozone regimes for ozone daily maxima averaged over the summer 2018 for the 22 target cities used in [13]. (**Left**): ACT results trained on CHIMERE setup based on Passant-GENEMIS. (**Right**): ACT results trained on CHIMERE setup based on TNO-SPEC-TEMP.

There is a very noticeable shift towards TRA-sensitive regimes replacing the 40% of cases classified as "Both IND and TRA" sensitive in the ACT version trained with the Passant-GENEMIS configuration. This is because industrial emissions are generally dominated by NMVOCs rather than $NO_x$. The significant increase in NMVOC concentrations in the TNO-SPEC-TEMP case tends to push towards $NO_x$-limited regimes and, therefore, a less significant impact of industrial emissions on $O_3$ formation.

## 6. Discussion and Conclusions

Here, we extend the study started in [13], by investigating how sensitive the ozone chemical regimes are for modelling parameterisations. Sensitivity tests were performed on two main aspects: increasing model resolution (from 25 km to 4 km) and updating emissions-related parametrisation. The conclusions are as follows:

- The main impact of increasing the resolution of the model is to concentrate emissions close to the emission release areas. This leads to an increase in ozone precursor emissions and, therefore, an increase in ozone production but also to higher consumption of ozone by reaction with NO (more titration impact). Therefore, depending on the production/destruction balance, the impact will be different. When the ozone indicator is representative of daily ozone peaks, increasing the resolution will have an impact on the ozone simulated in the reference simulation (without emission reductions) but relatively little impact on the ozone response to a given emission reduction. On the other hand, when focusing on the average ozone value (i.e., daily average ozone over the summer period), the resolution has a strong impact both on the initial ozone concentration values (without emission reductions) and on the ozone response. For the daily average indicator, we observed a switch to a titration regime for all the areas with significant $NO_x$ emissions and a reduction in the amplitude of the ozone response elsewhere;

- In the domain of interest in this study (i.e., the south of France), the impact of pollution imported from outside the simulation domain was studied by comparing simulations for which emission reductions were limited to the domain or applied to the whole of Europe. The results show that this external pollution did not affect the chemical regime which remains unchanged in response to changing boundary conditions. The first lever for action on ozone peaks remains the reduction of local and regional emissions, but in order to achieve higher levels of reduction, it is necessary to act at the European level to reduce ozone imports. However, this ozone import can only be effectively reduced by high levels of emission reduction (>70%). With such emission reductions, ozone import can be almost as effective as regional reductions to reduce ozone peaks. It is important to note that the study area (south-east France) is a zone with high ozone precursor emissions and strong photochemical activity. The efficiency of local emission reductions in other parts of Europe may be lower, and the contribution of European reductions higher;

- A change in the emission parameterisations may have a significant impact. Here, the implementation of the TNO-SPEC-TEMP emission parameterisation leads to non-negligible changes in the quantities of diurnal ozone precursors emitted in summer with an increase in the quantities of NMVOCs and a decrease in $NO_x$. The rise in NMVOC concentrations causes a shift towards a more $NO_x$-limited regime. For cities that were already in a $NO_x$-limited regime, the impact is limited, but in the opposite case (mainly northern European cities with high $NO_x$ levels often in titration regime), the shift induces a more balanced regime which makes $NO_x$ emission reduction more effective. It also results in greater sensitivity to reductions in traffic emissions than to reductions in industrial emissions;

- It is interesting to note that just because initial ozone levels are strongly modified does not mean that the ozone response to emission reductions will be, and vice versa. The impact on the ozone response will be strong if a regime change occurs with the parametrisation change. In particular, the sensitivity tests carried out showed that the

major differences were more likely to be simulated in cities with high NO$_x$ emissions and limited radiation. These cities are generally NMVOC-limited and/or in a titration regime and are particularly sensitive to changes in model parameters;

-    Finally, to complete the study, in addition to the sensitivity analysis, we studied the impact of traffic and industrial emission reductions (from 0 to 100%) on daily threshold exceedances in south-eastern France using the high-resolution model. Emission reductions in the road sector will make it possible to reduce exceedances of the ozone threshold value for the information of the public (180 µg/m$^3$) by a ratio of around 1:1 up to 60% reductions in emissions. The effectiveness of these reductions is lower when the ozone threshold is lower. Reductions in industrial emissions are less effective in both cases but are still worthwhile when the study area is the south-east of France.

Whether the conclusions in [13] are still valid can be answered as follows:

(1)    This study confirms the differences in the magnitude but also sometimes the sign of the O$_3$ response to emission reductions for the same city depending on the indicator chosen. Here, the impact on the daily average is compared to that on the daily peaks and the conclusions on the efficiency or counterproductivity of emission reduction measures are very different. However, the most significant changes are simulated for the daily average, which is not the most suitable indicator for studying the impact of ozone on health or vegetation;

(2)    The counterproductivity of abatement measures in areas with high emissions and low radiation is increased when the model resolution becomes finer but, on the other hand, the implementation of a new emissions scheme leads to a more balanced regime and greater efficiency of those reductions. In any case, these regions (mainly the large cities in the northern half of Europe) are particularly sensitive to the model's parameters and the chemical regime can be completely modified according to these parameters;

(3)    The conclusion that outside the titration regime, most cases show a higher sensitivity to emission reductions from traffic or equal sensitivity to emission reductions from traffic and industry is still valid after our sensitivity study;

(4)    Finally, sensitivity tests validate one of our previous conclusions on the limited effect of emission reduction measures when the O$_3$ indicator is an average over a long period (summer here). On the other hand, we were able to show that this effect is more important on peak ozone. While we only focus here on long-term emission reduction, we also show that mitigation measures can significantly reduce ozone peaks.

We conclude on the relevance and robustness of the innovative modelling framework introduced in [13] to infer relevant diagnostics on the efficiency of emission reduction efficiency to mitigate ozone air pollution. As expected, the spatial resolution and the representation of NMVOC emissions are important factors, which can change the assessment of the main ozone regimes, in some cases without changing the overall conclusions of the previous assessment.

These findings are illustrative of the modelling uncertainty, which is particularly important for cities with high NO$_x$ emissions and little solar radiation. To further increase the robustness of the assessment, it would be worthwhile to rely on an ensemble approach, involving a variety of chemistry-transport models and emission inventories at regional to local scales. Two important parameters that were not investigated in our sensitivity study, and which may also have a significant impact, are meteorological parameters and biogenic emissions. The research community has a role to play in better quantifying these uncertainties. But it is also important that any use of air quality modelling to support policy takes into account these factors. As far as Europe is concerned, the use of air quality modelling in the implementation of the European Directives is supported by the FAIRMODE network. This network launched an intercomparison platform ([42]), which is designed to address the issue of the sensitivity of model responses to emission changes, in particular, to assess, discuss, explain and minimise model discrepancies. Such initiatives are instrumental for fostering research that is tailored to ultimately address policy expectations.

**Author Contributions:** Conceptualisation, E.R., A.M. and A.C. (Augustin Colette); Methodology, E.R. and F.C.; Software, E.R., F.C. and A.C. (Adrien Chantreux); Supervision, G.V., A.M. and A.C. (Augustin Colette); Writing—original draft, E.R.; Writing—review and editing, E.R., F.C., A.C. (Adrien Chantreux), A.M. and A.C. (Augustin Colette). All authors have read and agreed to the published version of the manuscript.

**Funding:** Real reports financial support was provided by Concawe.

**Institutional Review Board Statement:** Not applicable.

**Informed Consent Statement:** Not applicable.

**Data Availability Statement:** Data will be made available on request. The data are not publicly available due privacy.

**Conflicts of Interest:** The authors declare the following financial interest/personal relationships which may be considered as potential competing interests: real reports financial support was provided by Concawe. The funder was not involved in the study design, collection, analysis, interpretation of data, the writing of this article or the decision to submit it for publication.

## Abbreviations

| Abbreviation | Meaning |
|---|---|
| $NO_x$ | nitrogen oxides |
| $O_3$ | ozone |
| VOC | Volatile Organic Compounds |
| ACT | Air Control Toolbox |
| $SOMO_3 5$ | Annual $O_3$ indicators calculated as the annual sum of daily means over 35 ppbv |
| NMVOC | Non-Methane Volatile Organic Compounds |
| CTM | Chemistry-Transport Model |
| ECMWF | European Centre for Medium Range Weather Forecasts |
| CEIP | Centre for Emission Inventories and Projections |
| INERIS | French National Institute for Industrial Environment and Risks |
| CAMS | Copernicus Atmosphere Monitoring Service |
| OHP | Observatoire de Haute Provence |
| MDA8 | Mean value of the daily maximum 8 h average |
| P72.8 | percentile 72.8 |
| EU | European Union |
| TNO | Nederlandse Organisatie voor Toegepast Natuurwetenschappelijk Onderzoek |
| LTO | Long Terme Objective |

## Appendix A

In the model, the monthly profiles used for emissions are specified by country, by pollutant and by emissions sector. The profiles used for France are detailed here as an example. For this specific figure, to be able to compare a single profile per pollutant and per country, it was decided to construct a combined profile, for which the profile for each sector is weighted by the emissions of the pollutant for that sector. For example, for NMVOC, the temporal profile of the solvents sector will carry more weight than that of aviation. Combined profiles were constructed with the original temporal profiles of CHIMERE (Passant-GENEMIS) and for the new TNO temporal profiles (TNO-TEMP). These emissions-weighted monthly profiles for NMVOCs and $NO_x$ are shown in Figures A1 and A2. The 2018 emissions for France are shown in Table A1.

**Table A1.** $NO_x$ and NMVOC emissions for France for the year 2019.

| GNFR13 Sector | $NO_x$ Emissions (kT) | NMVOC Emissions (kT) |
|---|---|---|
| A_PublicPower | 26 | 1 |
| B_Industry | 102 | 64 |

**Table A1.** *Cont.*

| GNFR13 Sector | NO$_x$ Emissions (kT) | NMVOC Emissions (kT) |
|---|---|---|
| C_OtherStationaryComb | 78 | 106 |
| D_Fugitive | 2 | 21 |
| E_Solvents | 1 | 302 |
| F_RoadTransport | 421 | 47 |
| G_Shipping | 11 | 9 |
| H_Aviation | 10 | 1 |
| I_Offroad | 84 | 21 |
| J_Waste | 2 | 8 |
| K_AgriLivestock | 10 | 196 |
| L_AgriOther | 64 | 202 |

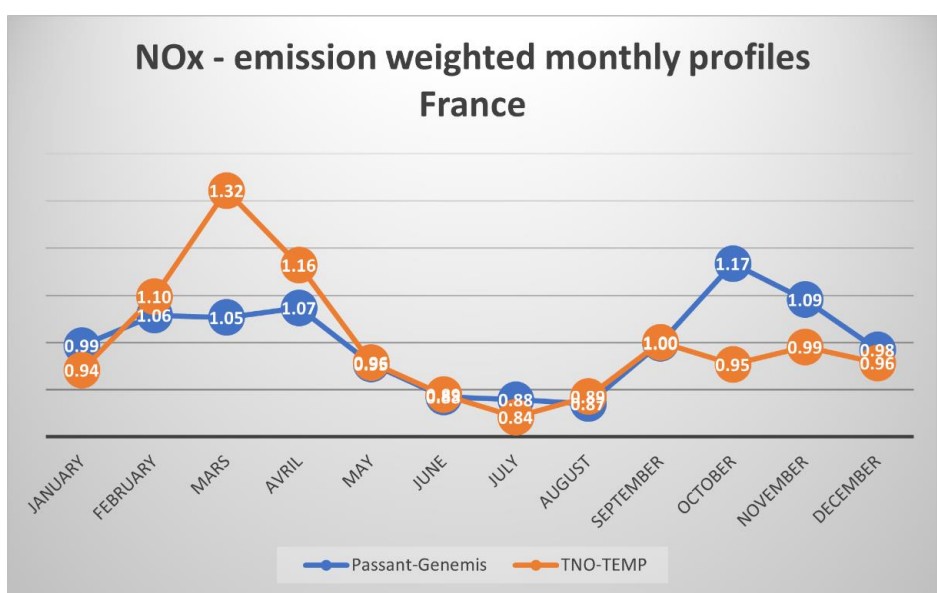

**Figure A1.** Emission-weighted monthly profiles for NO$_x$ as used for France in CHIMERE.

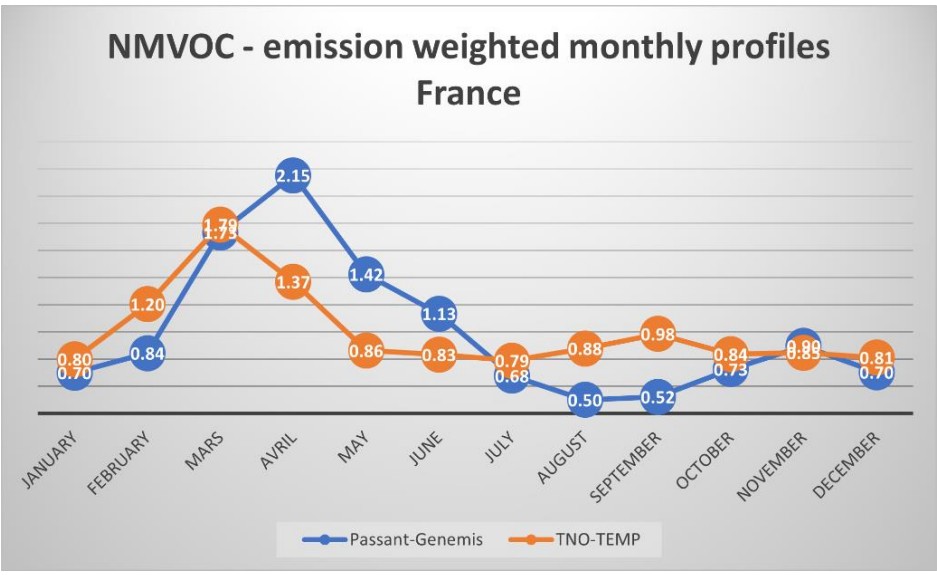

**Figure A2.** Emission-weighted monthly profiles for NMVOCs as used for France in CHIMERE.

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
