# Peer review of "Assessing the Robustness of Ozone Chemical Regimes to Chemistry-Transport Model Configurations"

_atmosphere, doi:10.3390/atmos15050532_

Round 1
Reviewer 1 Report
Comments and Suggestions for Authors
Report on manuscript atmosphere-2885841
“Assessing the robustness of ozone chemical regimes to chemistry-transport model configurations”
This manuscript discusses the sensitivity of the ozone chemical regimes to the model parameterizations as a follow-up of previous work. The manuscript is well-written and presented in a clear and correct language. The authors describe their method very well and presented their results. I think the paper can be accepted after minor revision. I would like the authors to address the following points:
- I would avoid adding references in the abstract. I think the abstract should deliver the message of the paper with no direct help from external references.
- Please discuss similar studies, and how your findings compare and add to the existing knowledge.
- I recommend adding an abbreviation table at the end of the manuscript (as an appendix)
- You refer to your previous paper, which is good, but I think you over-cited that work in this paper. I counted that you cited your previous paper (25 times). I think that can be reduced and kept in a few places where necessary.
Reviewer 2 Report
Comments and Suggestions for Authors
Journal: Atmosphere (ISSN 2073-4433)
Manuscript ID: atmosphere-2885841
Title: Assessing the robustness of ozone chemical regimes to chemistry-transport model configurations
General Comments
This reviewer's comment highlights the significance of the manuscript, which focuses on evaluating the impact of traffic and industrial emissions on ozone levels in various European cities. The study uses the Air Control Toolbox surrogate model to perform sensitivity analyses on model parameters, emphasizing the importance of assessing robustness in such investigations. The findings of this study are particularly noteworthy, as they reveal that increasing the model has a limited effect on ozone concentration peaks but significantly alters the response of ozone daily mean with a shift towards a titration regime in areas with substantial nitrogen oxides (NOx) emissions. Furthermore, the research sheds light on the importance of addressing pollution imported from outside the simulation domain and suggests that local and regional emission reductions remain the primary focus for ozone peak management. The manuscript also explores more recent temporal profiles and sectoral emission speciation, demonstrating a shift towards a more NOx-limited regime in several cities. Overall, the sensitivity tests conducted in this study emphasize that most differences are observed in cities with high NOx emissions and low solar radiation, reinforcing the initial conclusions drawn from the research. In summary, this manuscript provides valuable insights into the relationship between various emissions and ozone metrics in European cities, contributing to the understanding of effective strategies for ozone peak management and emission reduction.
Specific Comments:
1. Introduction: The introduction could benefit from providing a clearer context on the importance of studying ozone metrics and their link to traffic and industrial emissions. It would also be helpful to mention any existing research gaps that this study aims to address.
2. Methodology: While the use of the Air Control Toolbox surrogate model is mentioned, it would be beneficial to provide a brief explanation of the model and its relevance to the study. Additionally, more details on the sensitivity analyses performed and the rationale behind selecting specific parameters could be useful for the reader.
3. Results: The results section could be improved by including visual aids, such as graphs or charts, to better illustrate the impact of varying model parameters on ozone metrics. This would make it easier for readers to understand and compare the findings.
4. Discussion: The discussion could be enriched by including a comparison with previous studies and their findings. This would help establish the novelty and significance of the current research. Furthermore, it would be valuable to elaborate on the implications of the study's findings for policymakers and environmental management strategies.
5. Conclusion: The conclusion should be more concise and summarize the key findings and their implications more explicitly. It would also be helpful to mention any future research directions that could build upon the current study's findings.
6- References: There are fewer numbers of new references or recent references especially related to chemistry-transport model studies. Therefore, for current study should be given a strong impact if you can cite the following reference.
- Black Carbon Emissions from Traffic Contribute Sustainability to Air Pollution in Urban Cities of India
- Characterization of PAHs and n-alkanes in atmospheric aerosol of Jamshedpur City, India
- Emission sources, Characteristics and risk assessment of particulate bound Polycyclic Aromatic Hydrocarbons (PAHs) from traffic sites
- Characterization of PM10 over urban and rural sites of Rajnandgaon, central India
- Understanding Sources and Composition of Black Carbon and PM2.5 in Urban Environments in East India
7. Language and Clarity: Throughout the manuscript, the use of technical terms and abbreviations should be explained when first introduced. The overall language and clarity could be improved by revising the sentence structure and ensuring that the text is easily understandable for a broader audience.
8 The comma and full stops must be checked in the result section.
9 Improve the pixel of figures.
Comments on the Quality of English Language
Given in above authors comments
Reviewer 3 Report
Comments and Suggestions for Authors
The authors address a topical subject such as air pollution with ozone. Through this study, the authors evaluate a range of emissions reduction effects using the ACT surrogate model based on complete CHIMERE simulations. The study presented in this manuscript is based on another study carried out by the authors, and the results presented now are compared with the previous ones.
The research results confirm the research problem, but the background information doesn't provide enough needed to understand the results.
- Usually do not cite references-most of your abstract will describe what you have studied in your research what you have found and what you argue in your paper.
- The abstract must be redone, it is not well understood what was intended to be studied. The first sentence should introduce the topic under study and what are the various causes and interrelationships for this trend. Then, the following sentences should present the data, research, and analytical methods used in this study. The final sentences should address the main findings of the study and the implications and significance of this study.
- The text from lines 53-67 should be moved to the beginning of the Discussion section.
- In the Introduction, the part specifying what this study entails is very large, many details should be entered at the beginning of the Results or Materials section when specifying the analysis area.
- A numerical experiment description should be made, regarding the chemical mechanism, simulation domain, meteorological data (wind flow), initial conditions, and the numerical model and initial data.
- There is no information about the input data or the mechanisms used, there is a lot of talk, but the concrete data are missing. the results are directly given.
Round 2
Reviewer 2 Report
Comments and Suggestions for Authors
ISSN 2073-4433)
atmosphere-2885841
Assessing the robustness of ozone chemical regimes to chemistry-transport model configurations
1. Introduction:
- Briefly explain the significance of studying ozone metrics and the importance of understanding the impact of traffic and industrial emissions reduction on them.
2. Methodology:
- Provide a clearer explanation of the Air Control Toolbox surrogate model and its relevance to the study.
- Specify the cities and time period considered in the research.
3. Model Parametrisation Sensitivity Analyses:
- Elaborate on the various parameters analyzed and their relevance to the study's objectives.
- Discuss the methodology used for conducting these sensitivity analyses.
4. Results:
- Clarify the relationship between model sensitivity and ozone response to emissions changes, focusing on concentration peaks and daily mean values.
- Explain the concept of "titration regime" and its implications for ozone levels in zones with significant NOx emissions.
- Describe the impact of pollution imported from outside the simulation domain and its implications for ozone peaks reduction strategies.
5. European-level Action:
- Emphasize the importance of coordinated efforts at the European level to achieve higher reduction levels in ozone peaks.
6. Up-to-date Temporal Profiles and Emission Speciation:
- Explain the rationale behind exploring more recent temporal profiles and sectoral emission speciation.
- Highlight the key findings from these explorations, including the shift towards a more NOx-limited regime in some cities.
7. Overall Conclusions:
- Summarize the main findings and their implications for understanding ozone metrics and emission reduction strategies.
- Mention any limitations of the study and potential areas for future research.
8. References: There are fewer numbers of new references or recent references especially related to air quality/ chemistry-transport model studies. Therefore, for current study should be given a strong impact if you can cite the following reference.
- Black Carbon Emissions from Traffic Contribute Sustainability to Air Pollution in Urban Cities of India
- Characterization of PAHs and n-alkanes in atmospheric aerosol of Jamshedpur City, India
- Emission sources, Characteristics and risk assessment of particulate bound Polycyclic Aromatic Hydrocarbons (PAHs) from traffic sites
- Characterization of PM10 over urban and rural sites of Rajnandgaon, central India
- Understanding Sources and Composition of Black Carbon and PM2.5 in Urban Environments in East India
9. The comma and full stops must be checked in the result section.
10 Improve the pixel of figures.
Reviewer 3 Report
Comments and Suggestions for Authors
The manuscript can be published in the present form after the observations received by the authors.
Author Response
Thank you for your review. Unless I'm mistaken, I haven't seen any new questions/comments, other than those we answered during the first review phase.